**1. TITLE PAGE**
**Seasonal variability in methane and nitrous oxide fluxes from tropical peatlands in the**
**Western Amazon basin**
Teh, Yit Arn[1*], Murphy, Wayne A.[2], Berrio, Juan-Carlos[2], Boom, Arnound[2], and Page, Susan E.[2]
[1] Institute of Biological and Environmental Sciences, University of Aberdeen
[2] Department of Geography, University of Leicester
* Author to whom all correspondence should be addressed; email: yateh@abdn.ac.uk
**2. ABSTRACT**
The Amazon plays a critical role in global atmospheric budgets of methane ($CH_4$) and nitrous
oxide ($N_2O$). However, while we have a relatively good understanding of the continental-scale
flux of these greenhouse gases (GHGs), one of the key gaps in knowledge is the specific
contribution of peatland ecosystems to the regional budgets of these GHGs. Here we report
$CH_4$ and $N_2O$ fluxes from lowland tropical peatlands in the Pastaza-Marañón foreland basin
(PMFB) in Peru, one of the largest peatland complexes in the Amazon basin. The goal of this
research was to: quantify the range and magnitude of $CH_4$ and $N_2O$ fluxes from this region;
assess seasonal trends in trace gas exchange; and determine the role of different
environmental variables in driving GHG flux. Trace gas fluxes were determined from the most
numerically-dominant peatland vegetation types in the region: forested vegetation, forested
(short pole) vegetation, *Mauritia flexuosa*-dominated palm swamp, and mixed palm swamp.
Data were collected in both wet and dry seasons over the course of four field campaigns from
2012 to 2014. Diffusive $CH_4$ emissions averaged $36.05 \pm 3.09$ mg $CH_4$-C m$^{-2}$ d$^{-1}$ across the
entire dataset, with diffusive $CH_4$ flux varying significantly among vegetation types and
between seasons. Net ebullition of $CH_4$ averaged $973.3 \pm 161.4$ mg $CH_4$-C m$^{-2}$ d$^{-1}$, and did not
vary significantly among vegetation types nor between seasons. Diffusive $CH_4$ flux was
greatest for mixed palm swamp ($52.0 \pm 16.0$ mg $CH_4$-C m$^{-2}$ d$^{-1}$), followed by *M. flexuosa* palm
swamp ($36.7 \pm 3.9$ mg $CH_4$-Cm$^{-2}$ d$^{-1}$), forested (short pole) vegetation ($31.6 \pm 6.6$ mg $CH_4$-Cm$^{-2}$
d$^{-1}$), and forested vegetation ($29.8 \pm 10.0$ mg $CH_4$-C m$^{-2}$ d$^{-1}$). Diffusive $CH_4$ flux also showed
marked seasonality, with divergent seasonal patterns among ecosystems. Forested
vegetation and mixed palm swamp showed significantly higher dry season ($47.2 \pm 5.4$ mg $CH_4$-
C m$^{-2}$ d$^{-1}$ and $85.5 \pm 26.4$ mg $CH_4$-C m$^{-2}$ d$^{-1}$, respectively) compared to wet season emissions
(6.8 ± 1.0 mg $CH_4$-C $m^{-2}d^{-1}$ and 5.2 ± 2.7 mg $CH_4$-C $m^{-2}d^{-1}$, respectively). In contrast, forested
(short pole) vegetation and *M. flexuosa* palm swamp showed the opposite trend, with dry
season flux of 9.6 ± 2.6 and 25.5 ± 2.9 mg $CH_4$-C $m^{-2}d^{-1}$, respectively, versus wet season flux
of 103.4 ± 13.6 and 53.4 ± 9.8 mg $CH_4$-C $m^{-2}d^{-1}$, respectively. These divergent seasonal trends
may be linked to very high water tables (>1 m) in forested vegetation and mixed palm swamp
during the wet season, which may have constrained $CH_4$ transport across the soil-atmosphere
interface. Diffusive $N_2O$ flux was very low (0.70 ± 0.34 μg $N_2O$-N $m^{-2}d^{-1}$), and did not vary
significantly among ecosystems nor between seasons. We conclude that peatlands in the
PMFB are large and regionally significant sources of atmospheric $CH_4$, that need to be better
accounted for in regional emissions inventories. In contrast, $N_2O$ flux was negligible,
suggesting that this region does not make a significant contribution to regional atmospheric
budgets of $N_2O$. The divergent seasonal pattern in $CH_4$ flux among vegetation types challenges
our underlying assumptions of the controls on $CH_4$ flux in tropical peatlands, and emphasizes
the need for more process-based measurements during high water table periods.
**KEYWORDS**
methane, nitrous oxide, peat, tropical peatland, Amazonia, Peru

## 3. INTRODUCTION

The Amazon basin plays a critical role in the global atmospheric budgets of carbon (C) and greenhouse gases (GHGs) such as methane ($CH_4$) and nitrous oxide ($N_2O$). Recent basin-wide studies suggest that the Amazon as a whole accounts for approximately 7 % of global atmospheric $CH_4$ emissions (Wilson et al., 2016). $N_2O$ emissions are of a similar magnitude, with emissions ranging from 2-3 Tg $N_2O$-N year$^{-1}$ (or, approximately 12-18 % of global atmospheric emissions) (Huang et al., 2008;Saikawa et al., 2014;Saikawa et al., 2013). While we have a relatively strong understanding of the role that the Amazon plays in regional and global atmospheric budgets of these gases, one of the key gaps in knowledge is the contribution of specific ecosystem types to regional fluxes of GHGs (Huang et al., 2008;Saikawa et al., 2014;Saikawa et al., 2013). In particular, our understanding of the contribution of Amazonian wetlands to regional C and GHG budgets is weak, as the majority of past ecosystem-scale studies have focused on *terra firme* forests and savannas (D'Amelio et al., 2009;Saikawa et al., 2013;Wilson et al., 2016;Kirschke et al., 2013;Nisbet et al., 2014). Empirical studies of GHG fluxes from Amazonian wetlands are more limited in geographic scope and have focused on three major areas: wetlands in the state of Amazonas near the city of Manaus (Devol et al., 1990;Bartlett et al., 1990;Bartlett et al., 1988;Keller et al., 1986), the Pantanal region (Melack et al., 2004;Marani and Alvalá, 2007;Liengaard et al., 2013), and the Orinoco River basin (Smith et al., 2000;Lavelle et al., 2014). Critically, none of the ecosystems sampled in the past were peat-forming ones; rather, the habitats investigated were non-peat forming (i.e. mineral or organo-mineral soils), seasonally-inundated floodplain forests (i.e. *varzea*), rivers or lakes.

Peatlands are one of the major wetland habitats absent from current bottom-up GHG
inventories for the Amazon basin, and are often grouped together with non-peat forming
wetlands in regional atmospheric budgets (Wilson et al., 2016). Unlike their Southeast Asian
counterparts, most peatlands in the Amazon basin are unaffected by human activity at the
current time (Lahteenoja et al., 2009a; Lahateenoja et al. 2009b; Lahteenoja and Page 2011),
except for ecosystems in the Madre de Dios region in southeastern Peru, which are impacted
by gold mining (Householder et al., 2012). Because we have little or no data on ecosystem-
level land-atmosphere fluxes from Amazonian peatlands (Lahteenoja et al., 2012;Lahteenoja
et al., 2009b;Kirschke et al., 2013;Nisbet et al., 2014), it is difficult to ascertain if rates of GHG
flux from these ecosystems are similar to or different from mineral soil wetlands (e.g. *varzea*).
Given that underlying differences in plant community composition and soil properties are
known to modulate the cycling and flux of GHGs in wetlands (Limpens et al., 2008;Melton et
al., 2013;Belyea and Baird, 2006;Sjögersten et al., 2014), expanding our observations to
include a wider range of wetland habitats is critical in order to improve our understanding of
regional trace gas exchange, and also to determine if aggregating peat and mineral soil
wetlands together in bottom-up emissions inventories are appropriate for regional budget
calculations. Moreover, Amazonian peatlands are thought to account for a substantial land
area (i.e. up to 150,000 km$^2$) (Schulman et al., 1999;Lahteenoja et al., 2012), and any
differences in biogeochemistry among peat and mineral/organo-mineral soil wetlands may
therefore have important implications for understanding and modelling the biogeochemical
functioning of the Amazon basin as a whole.

Since the identification of extensive peat forming wetlands in the north (Lahteenoja et al.,
2009a; Lahateenoja et al. 2009b; Lahteenoja and Page 2011) and south (Householder et al.,
2012) of the Peruvian Amazon, several studies have been undertaken to better characterize
these habitats, investigating vegetation composition and habitat diversity (Draper et al., 2014;
Kelly et al., 2014; Householder et al., 2012; Lahteenoja and Page, 2011), vegetation history
(Lahteenoja and Roucoux et al., 2010), C stocks (Lahteenoja et al., 2012; Draper et al., 2014),
hydrology (Kelly et al., 2014), and peat chemistry (Lahteenoja et al., 2009a; Lahteenoja et al.,
2009b). Most of the studies have focused on the Pastaza-Marañón foreland basin (PMFB),
where one of the largest stretches of contiguous peatlands have been found (Lahteenoja et
al 2009a; Lahteenoja and Page, 2011; Kelly et al, 2014), covering an estimated area of 35,600
± 2,133 km$^2$ (Draper et al., 2014). Up to 90% of the peatlands in the PMFB lie in flooded
backwater river margins on floodplains and are influenced by large, annual fluctuations in
water table caused by the Amazonian flood pulse (Householder et al., 2012;Lahteenoja et al.,
2009a). These floodplain systems are dominated by peat deposits that range in depth from
~3.9 m (Lahteenoja et al., 2009a) to ~12.9 m (Householder et al., 2012). The remaining 10%
of these peatlands are not directly influenced by river flow and form domed (i.e. raised)
nutrient-poor bogs that likely only receive water and nutrients from rainfall (Lahteenoja et
al., 2009b). These nutrient-poor bogs are dominated by large, C-rich forests (termed "pole
forests"), that represent a very high density C store (total pool size of 1391 ± 710 Mg C ha$^{-1}$,
which includes both above- and belowground stocks); exceeding in fact the C density of
nearby floodplain systems (Draper et al., 2014). Even though the peats in these nutrient-poor
bogs have a relatively high hydraulic conductivity, they act as natural stores of water because
of high rainwater inputs (>3000 mm per annum), which help to maintain high water tables,
even during parts of the dry season (Kelly et al., 2014).

$CH_4$ flux in tropical soils are regulated by the complex interplay among multiple factors that
regulate $CH_4$ production, oxidation, and transport. Key factors include: redox/water table
depth (Couwenberg et al., 2010;Couwenberg et al., 2011;Silver et al., 1999;Teh et al.,
2005;von Fischer and Hedin, 2007), plant productivity (von Fischer and Hedin, 2007;Whiting
and Chanton, 1993), soil organic matter lability (Wright et al., 2011), competition for C
substrates among anaerobes (Teh et al., 2008;Teh and Silver, 2006;von Fischer and Hedin,
2007), and presence of plants capable of facilitating atmospheric egress (Pangala et al., 2013).
Of all these factors, fluctuation in soil redox conditions, as mediated by variations in water
table depth, is perhaps most critical in regulating $CH_4$ dynamics (Couwenberg et al.,
2010;Couwenberg et al., 2011), because of the underlying physiology of the microbes that
produce and consume $CH_4$. Methanogenic archaea are obligate anaerobes that only produce
$CH_4$ under anoxic conditions (Conrad, 1996); as a consequence, they are only active in stably
anoxic soil microsites or soil layers, where they are protected from the effects of strong
oxidants such as oxygen or where competition for reducing equivalents (e.g. acetate, $H_2$) from
other anaerobic microorganisms is eliminated (Teh et al., 2008;Teh and Silver, 2006;Teh et
al., 2005;von Fischer and Hedin, 2002;von Fischer and Hedin, 2007). $CH_4$ oxidation, on the
other hand, is thought to be driven primarily by aerobic methanotrophic bacteria in tropical
soils (Hanson and Hanson, 1996;Teh et al., 2005;Teh et al., 2006;von Fischer and Hedin,
2002;von Fischer and Hedin, 2007), with anaerobic $CH_4$ oxidation playing a quantitatively
smaller role (Blazewicz et al., 2012). Thus, fluctuations in redox or water table depth play a
fundamental role in directing the flow of C among different anaerobic pathways (Teh et al.,
2008;Teh and Silver, 2006;von Fischer and Hedin, 2007), and shifting the balance between
production and consumption of $CH_4$ (Teh et al., 2005;von Fischer and Hedin, 2002). Moreover,
water table or soil moisture fluctuations are also thought to profoundly influence $CH_4$
transport dynamics throughout the soil profile, changing the relative partitioning of $CH_4$
among different transport pathways such as diffusion, ebullition, and plant-facilitated
transport (Whalen, 2005;Jungkunst and Fiedler, 2007).

Controls on $N_2O$ flux are also highly complex (Groffman et al., 2009), with $N_2O$ originating
from as many as four separate sources (e.g. bacterial ammonia oxidation, archaeal ammonia
oxidation, denitrification, dissimilatory nitrate reduction to ammonium), each with different
environmental controls (Baggs, 2008;Morley and Baggs, 2010;Firestone and Davidson,
1989;Firestone et al., 1980;Pett-Ridge et al., 2013;Silver et al., 2001;Prosser and Nicol, 2008).
Key factors regulating soil $N_2O$ flux include: redox, soil moisture content or water table depth,
temperature, pH, labile C availability, and labile N availability (Groffman et al., 2009). As is the
case for $CH_4$, variations in redox/water table depth plays an especially prominent role in
regulating $N_2O$ flux in tropical peatland ecosystems, because all of the processes that produce
$N_2O$ are redox-sensitive, with bacterial or archaeal ammonia oxidation occurring under
aerobic conditions (Prosser and Nicol, 2008;Firestone and Davidson, 1989;Firestone et al.,
1980) whereas nitrate-reducing processes (i.e. denitrification, dissimilatory nitrate reduction
to ammonium) are anaerobic ones (Firestone and Davidson, 1989;Firestone et al.,
1980;Morley and Baggs, 2010;Silver et al., 2001). Moreover, for nitrate reducing processes,
which are believed to be the dominant source of $N_2O$ in wet systems, the extent of
anaerobiosis also controls the relative proportion of $N_2O$ or $N_2$ produced during dissimilatory
metabolism (Firestone and Davidson, 1989;Firestone et al., 1980;Morley and Baggs,
2010;Silver et al., 2001).

In order to improve our understanding of the biogeochemistry and rates of GHG exchange
from Amazonian peatlands, we conducted a preliminary study of $CH_4$ and $N_2O$ fluxes from
forested peatlands in the PMFB. The main objectives of this are to:
1. Quantify the magnitude and range of soil $CH_4$ and $N_2O$ fluxes from a sub-set of

174        peatlands in the PMFB that represent dominant vegetation types

2. Determine seasonal patterns of trace gas exchange
3. Establish the relationship between trace gas fluxes and environmental variables
Sampling was concentrated on the four most dominant vegetation types in the area, based
on prior work by the investigators (Lahteenoja and Page, 2011). Trace gas fluxes were
captured from both floodplain systems and nutrient-poor bogs in order to account for
underlying differences in biogeochemistry that may arise from variations in hydrology.
Sampling was conducted during four field campaigns (two wet season, two dry season) over
a 27-month period, extending from February 2012 to May 2014.


**4. MATERIALS AND METHODS**
**4.1 Study site and sampling design**
The study was carried out in the lowland tropical peatland forests of the PMFB, between 2
and 35 km south of the city of Iquitos, Peru (Lahteenoja et al., 2009a; Lahteenoja et al., 2009b)
(Figure 1, Table 1). The mean annual temperature is 26 °C, annual precipitation is c. 3,100
mm, relative humidity ranges from 80-90 %, and altitude ranges from c. 90 to 130 m above
sea level (Marengo 1998). The northwestern Amazon basin near Iquitos experiences
pronounced seasonality, which is characterized by consistently high annual temperatures, but
marked seasonal variation in precipitation (Tian et al., 1998), and an annual river flood pulse
linked to seasonal discharge from the Andes (Junk et al., 1989). Precipitation events are
frequent, intense and of significant duration during the wet season (November to May) and
infrequent, intense and of short duration during the dry season (June to August). September
and October represent a transitional period between dry and wet seasons, where rainfall
patterns are less predictable. Catchments in this region receive no less than 100 mm of rain
per month (Espinoza Villar et al., 2009a; Espinoza Villar et al., 2009b) and >3000 mm of rain
per year. River discharge varies by season, with the lowest discharge between the dry season
months of August and September. Peak discharge from the wet season flood pulse occurs
between April and May, as recorded at the Tamshiyaku River gauging station (Espinoza Villar
et al., 2009b).

Histosols form the dominant soil type for peatlands in this region (Andriesse, 1988;Lahteenoja
and Page, 2011). Study sites are broadly classified as nutrient-rich, intermediate, or nutrient-
poor (Lahteenoja and Page, 2011), with pH ranging from 3.5 to 7.2 (Lahteenoja and Page,
2011;Lahteenoja et al., 2009a;Lahteenoja et al., 2009b). More specific data on pH for our plots
are presented in Table 3. Nutrient-rich (i.e. minerotrophic) sites tend to occur on floodplains
and river margins, and account for at least 60 % of the peatland cover in the PMFB (Lahteenoja
and Page, 2011;Draper et al., 2014). They receive water, sediment, and nutrient inputs from
the annual Amazon river flood pulse (Householder et al., 2012;Lahteenoja and Page, 2011),
leading to higher inorganic nutrient content, of which Ca and other base cations form major
constituents (Lahteenoja and Page, 2011). Many of the soils in these nutrient-rich areas are
fluvaquentic Tropofibrists (Andriesse, 1988), and contain thick mineral layers or minerogenic
intrusions, reflective of episodic sedimentation events in the past (Lahteenoja and Page,
2011). In contrast, nutrient-poor (i.e. oligotrophic) sites tend to occur further in-land
(Lahteenoja and Page, 2011;Draper et al., 2014). They are almost entirely rain-fed, and
receive low or infrequent inputs of water and nutrients from streams and rivers (Lahteenoja
and Page, 2011). These ecosystems account for 10 to 40 % of peatland cover in the PMFB,
though precise estimates vary depending on the land classification scheme employed
(Lahteenoja and Page, 2011;Draper et al., 2014). Soil Ca and base cation concentrations are
significantly lower in these sites compared to nutrient-rich ones, with similar concentrations
to that of rainwater (Lahteenoja and Page, 2011). Soils are classified as typic or hydric
Tropofibrists (Andriesse, 1988). Even though Ca and base cations themselves play no direct
role in modulating $CH_4$ and $N_2O$ fluxes, underlying differences in soil fertility may indirectly
influence $CH_4$ and $N_2O$ flux by influencing the rate of labile C input to the soil, the
decomposability of organic matter, and the overall throughput of C and nutrients through the
plant-soil system (Firestone and Davidson, 1989;Groffman et al., 2009;von Fischer and Hedin,
2007;Whiting and Chanton, 1993).

We established 239 sampling plots (~30 m$^2$ per plot) within five tropical peatland sites that
captured four of the dominant vegetation types in the region (Draper et al.,
2014;Householder et al., 2012;Kelly et al., 2014;Lahteenoja and Page, 2011), and which
encompassed a range of nutrient availabilities (Figure 1, Table 1) (Lahteenoja and Page,
2011;Lahteenoja et al., 2009a). These four dominant vegetation types included: forested
vegetation (nutrient-rich; n= 21 plots), forested (short pole) vegetation (nutrient-poor; n= 47
plots), *Mauritia flexuosa*-dominated palm swamp (intermediate fertility, n= 153 plots), and
mixed palm swamp (nutrient-rich; n=18 plots) (Table 1).  Four of the study sites (Buena Vista,
Charo, Miraflores, and Quistococha) were dominated by only one vegetation type, whereas
San Jorge contained a mixture of *M. flexuosa* palm swamp and forested (short pole)
vegetation (Table 1). As a consequence, both vegetation types were sampled in San Jorge to
develop a more representative picture of GHG fluxes from this location. Sampling efforts were
partially constrained by issues of site access; some locations were difficult to access (e.g.
centre of the San Jorge peatland) due to water table height and navigability of river channels;
as a consequence, sampling patterns were somewhat uneven, with higher sampling densities
in some peatlands than in others (Table 1).

In each peatland site, transects were established from the edge of the peatland to its centre.
Each transect varied in length from 2 to 5 km, depending on the relative size of the peatland.
Randomly located sampling plots (~30 m$^2$ per plot) were established at 50 or 200 m intervals
along each transect, from which GHG fluxes and environmental variables were measured
concomitantly. The sampling interval (i.e. 50 or 200 m) was determined by the length of the
transect or size of the peatland, with shorter sampling intervals (50 m) for shorter transects
(i.e. smaller peatlands) and longer sampling intervals (200 m) for longer transects (i.e. larger
peatlands).

## 4.2 Quantifying soil-atmosphere exchange

Soil-atmosphere fluxes ($CH_4$, $N_2O$) were determined in four campaigns over a two-year annual
water cycle: February 2012 (wet season), June-August 2012 (dry season), June-July 2013 (dry
season), and May-June 2014 (wet season). The duration of the campaign for each study site
varied depending on its size. Each study site was generally sampled only once for each
campaign, except for a sub-set of plots within each vegetation type where diurnal studies
were conducted to determine if $CH_4$ and $N_2O$ fluxes varied over daily time steps. Gas exchange
was quantified using a floating static chamber approach (Livingston and Hutchinson, 1995;
Teh et al., 2011).  Static flux measurements were made by enclosing a 0.225 m$^2$ area with a
dark, single component, vented 10 L flux chamber. No chamber bases (collars) were used due
to the highly saturated nature of the soils. In most cases, a standing water table was present
at the soil surface, so chambers were placed directly onto the water. In the absence of a
standing water table, a weighted skirt was applied to create an airtight seal. Under these drier
conditions, chambers were placed carefully on the soil surface. In order to reduce the risk of
pressure-induced ebullition or disruption to soil gas concentration profiles caused by the
investigators' footfall, flux chambers were lowered from a distance of 2-m away using a 2-m
long pole. Gas samples were collected with syringes using >2 m lengths of Tygon® tubing,
after thoroughly purging the dead volumes in the sample lines. To promote even mixing
within the headspace, chambers were fitted with small computer fans (Pumpanen et al.,
2004). Headspace samples were collected from each flux chamber at five intervals over a 25
minute enclosure period using a gas tight syringe. Gas samples were stored in evacuated
Exetainers® (Labco Ltd., Lampeter UK), shipped to the UK, and subsequently analysed for $CH_4$,
$CO_2$ and $N_2O$ concentrations using Thermo TRACE GC Ultra (Thermo Fischer Scientific Inc.,
Waltham, Massachusetts, USA) at the University of St. Andrews. Chromatographic separation
was achieved using a Porapak-Q column, and gas concentrations determined using a flame
ionization detector (FID) for $CH_4$, a methanizer-FID for $CO_2$, and an electron capture detector
(ECD) for $N_2O$. Instrumental precision, determined from repeated analysis of standards, was
< 5% for all detectors.

Diffusive fluxes were determined by using the JMP IN version 11 (SAS Institute, Inc., Cary,
North Carolina, USA) statistical package to plot best-fit lines to the data for headspace
concentration against time for individual flux chambers, with fluxes calculated from linear or
non-linear regressions depending on the individual concentration trend against time (Teh et
al., 2014). Gas mixing ratios (ppm) were converted to areal fluxes by using the Ideal Gas Law
to solve for the quantity of gas in the headspace (on a mole or mass basis) and normalized by
the surface area of each static flux chamber (Livingston and Hutchinson, 1995). Ebullition-
derived $CH_4$ fluxes were also quantified in our chambers where evidence of ebullition was
found. This evidence consisted of either: (i) rapid, non-linear increases in $CH_4$ concentration
over time; (ii) abrupt, stochastic increases in $CH_4$ concentration over time; or (iii) an abrupt
stochastic increase in $CH_4$ concentration, followed by a linear decline in concentration. For
observations following pattern (i), flux was calculated by fitting a quadratic regression
equation to the data ($P < 0.05$), and $CH_4$ flux determined from the initial steep rise in $CH_4$
concentration. For data following pattern (ii), the ebullition rate was determined by
calculating the total $CH_4$ production over the course of the bubble event, in-line with prior
work conducted by the investigators (Teh et al., 2011). Last, for data following pattern (iii), a
best-fit line was plotted to the $CH_4$ concentration data after the bubble event, and a net rate
of $CH_4$ uptake calculated from the gradient of the line. While observations (i) – (iii) all reflect
the effects of ebullition, only observations following patterns (i) and (ii) indicate net emission
to the atmosphere, whereas observations following pattern (iii) indicate emission followed by
net uptake. As a consequence, patterns (i) and (ii) were categorized as "net ebullition" (i.e.
net efflux) whereas observations following pattern (iii) were categorized as "ebullition-driven
$CH_4$ uptake" (i.e. net influx).

**4.3 Environmental variables**
To investigate the effects of environmental variables on trace gas fluxes, we determined air
temperature, soil temperature, chamber headspace temperature, soil pH, soil electrical
conductivity (EC; $\mu Scm^{-2}$), dissolved oxygen concentration of the soil pore water (DO;
measured as percent saturation, %) in the top 15 cm of the peat column, and water table
position concomitant with gas sampling. Air temperature (measured 1.3 m above the soil)
and chamber headspace temperature were measured using a Checktemp® probe and meter
(Hanna Instruments LTD, Leighton Buzzard, UK). Peat temperature, pH, DO and EC were
measured at a depth of 15 cm below the peat surface and recorded *in situ* with each gas
sample using a HACH® rugged outdoor HQ30D multi meter and pH, LDO or EC probe. At sites
where the water level was above the peat surface, the water depth was measured using a
meter rule. Where the water table was at or below the peat surface, the water level was
measured by auguring a hole to 1 m depth and measuring water table depth using a meter
rule.

**4.4 Statistical Analyses**
Statistical analyses were performed using JMP IN version 11 (SAS Institute, Inc., Cary, North
Carolina, USA). Box-Cox transformations were applied where the data failed to meet the
assumptions of analysis of variance (ANOVA); otherwise, non-parametric tests were applied
(e.g. Wilcoxon signed-rank test). ANOVA and analysis of co-variance (ANCOVA) were used to
test for relationships between gas fluxes and vegetation type, season, and environmental
variables. When determining the effect of vegetation type on gas flux, data from different
study sites (e.g. San Jorge and Miraflores) were pooled together. Means comparisons were
tested using a Fisher's Least Significant Difference (LSD) test.


**5. RESULTS**
**5.1 Differences in gas fluxes and environmental variables among vegetation types**
All vegetation types were net sources of $CH_4$, with an overall mean (± standard error) diffusive
flux of 36.1 ± 3.1 mg $CH_4$-C $m^{-2} d^{-1}$ and a mean net ebullition flux of 973.3 ± 161.4 mg $CH_4$-C
$m^{-2} d^{-1}$ (Figure 2, Table 2). We also saw examples of ebullition-driven $CH_4$ uptake (i.e. a sudden
or stochastic increase in $CH_4$ concentration, followed immediately by a rapid linear decline in
concentration), with a mean rate of -504.1 ± 84.4 mg $CH_4$-C $m^{-2} d^{-1}$ (Table 2). Diffusive fluxes
of $CH_4$ accounted for the majority of observations (83.3 to 93.1 %), while ebullition fluxes
accounted for a much smaller proportion of observations (6.9 to 16.7 %; Table 2).

Diffusive $CH_4$ flux varied significantly among the four vegetation types sampled in this study
(two-way ANOVA with vegetation, season and their interaction, $F_{7, 979}$ = 13.2, $P<0.0001$; Fig.
2a). However, the effect of vegetation was relatively weak (see ANCOVA results in the section
'Relationships between gas fluxes and environmental variables'), and a means comparison
test on the pooled data was unable to determine which means differed significantly from the
others (Fisher's LSD, $P > 0.05$). For the pooled data, the overall numerical trend was that mixed
palm swamp showed the highest mean flux (52.0 ± 16.0 mg $CH_4$-C $m^{-2} d^{-1}$), followed by *M.*
*flexuosa* palm swamp (36.7 ± 3.9 mg $CH_4$-$Cm^{-2} d^{-1}$), forested (short pole) vegetation (31.6 ±
6.6 mg $CH_4$-$Cm^{-2} d^{-1}$), and forested vegetation (29.8 ± 10.0 mg $CH_4$-C $m^{-2} d^{-1}$). $CH_4$ ebullition
(i.e. net ebullition and ebullition-driven uptake) did not vary significant among vegetation
types nor between seasons (Table 2). Broadly speaking, however, we saw a greater frequency
of ebullition in the *M. flexuosa* palm swamp, followed by mixed palm swamp, forested
vegetation, and forested (short pole vegetation) (Table 2).

These study sites were also a weak net source of $N_2O$, with a mean diffusive flux of 0.70 ± 0.34
µg $N_2O$-N $m^{-2} d^{-1}$. We saw only limited evidence of ebullition of $N_2O$, with only three chambers
out of 1181 (0.3 % of observations) showing evidence of $N_2O$ ebullition. These data were
omitted from the analysis of diffusive flux of $N_2O$. Because of the high variance in diffusive
$N_2O$ flux among plots, analysis of variance indicated that mean diffusive $N_2O$ flux did not differ
significantly among vegetation types (two-way ANOVA, $P > 0.5$, Fig. 2b). However, when the
$N_2O$ flux data were grouped by vegetation type, we see that some vegetation types tended
to function as net atmospheric sources, while others acted as atmospheric sinks (Fig. 2b, Table
3). For example, the highest $N_2O$ emissions were observed from *M. flexuosa* palm swamp
$(1.11 \pm 0.44\ \mu g\ N_2O\text{-}N\ m^{-2}\ d^{-1})$ and forested vegetation $(0.20 \pm 0.95\ \mu g\ N_2O\text{-}N\ m^{-2}\ d^{-1})$. In
contrast, forested (short pole) vegetation and mixed palm swamp were weak sinks for $N_2O$,
with a mean flux of -0.01 ± 0.84 and $-0.21 \pm 0.70\ \mu g\ N_2O\text{-}N\ m^{-2}\ d^{-1}$, respectively.

Soil pH varied significantly among vegetation types (data pooled across all seasons; ANOVA,
$P < 0.0001$, Table 3). Multiple comparisons tests indicated that mean soil pH was significantly
different for each of the vegetation types (Fisher's LSD, $P < 0.0001$, Table 3), with the lowest
pH in forested (short pole) vegetation (4.10 ± 0.04), followed by *M. flexuosa* palm swamp
(5.32 ± 0.02), forested vegetation (6.15 ± 0.06), and the mixed palm swamp (6.58 ± 0.04).

Soil dissolved oxygen (DO) content varied significantly among vegetation types (data pooled
across all seasons; Kruskal-Wallis, $P < 0.0001$, Table 3). Multiple comparisons tests indicated
that mean DO was significantly different for each of the vegetation types (Fisher's LSD, $P <$
0.05, Table 3), with the highest DO in the forested (short pole) vegetation (25.2 ± 2.1 %),
followed by the *M. flexuosa* palm swamp (18.1 ± 1.0 %), forested vegetation (11.8 ± 2.8 %),
and the mixed palm swamp (0.0 ± 0.0 %).

Electrical conductivity (EC) varied significantly among vegetation types (data pooled across all
seasons; Kruskal-Wallis, $P < 0.0001$, Table 3). Multiple comparison tests indicated that mean
EC was significantly different for each of the vegetation types (Fisher's LSD, $P < 0.05$; Table 3),
with the highest EC in the mixed palm swamp ($170.9 \pm 6.0$ µs m$^{-2}$), followed by forested
vegetation ($77.1 \pm 4.2$ µs m$^{-2}$), *M. flexuosa* palm swamp ($49.7 \pm 1.4$ µs m$^{-2}$) and the forested
(short pole) vegetation ($40.9 \pm 3.5$ µs m$^{-2}$).

Soil temperature varied significantly among vegetation types (data pooled across all seasons;
ANOVA, $P < 0.0001$, Table 3). Multiple comparisons tests indicated that soil temperature in
forested (short pole) vegetation was significantly lower than in the other vegetation types
(Table 3); whereas the other vegetation types did not differ in temperature amongst
themselves (Fisher's LSD, $P < 0.05$, Table 3).

Air temperature varied significantly among vegetation types (data pooled across all seasons;
ANOVA, $P < 0.0001$, Table 3). Multiple comparisons tests indicated that air temperature in *M.*
*flexuosa* palm swamp was significantly lower than in the other vegetation types; whereas the
other vegetation types did not differ in temperature amongst themselves (Fisher's LSD, $P <$
$0.05$, Table 3).

Water table depths varied significantly among vegetation types (data pooled across all
seasons; ANOVA, $P < 0.0001$, Table 3). The highest mean water tables were observed in mixed
palm swamp ($59.6 \pm 9.3$ cm), followed by forested vegetation ($34.0 \pm 6.9$ cm), *M. flexuosa*
palm swamp (17.4 ± 1.2 cm), and forested (short pole) vegetation (3.5 ± 1.0 cm) (Fisher's LSD,
$P < 0.0005$).

**5.2 Temporal variations in gas fluxes and environmental variables**
The peatlands sampled in this study showed pronounced seasonal variability in diffusive $CH_4$
flux (two-way ANOVA, $F_{7, 979}$ = 13.2, $P<0.0001$; Table 4). For ebullition of $CH_4$ and ebullition-
driven uptake of $CH_4$, mean fluxes varied between seasons, but high variability meant that
these differences were not statistically significant ((two-way ANOVA, $P > 0.8$; Table 2).
Diffusive $N_2O$ flux showed no seasonal trends (two-way ANOVA, $P > 0.5$), and therefore will
not be discussed further here. Diurnal studies suggest that neither diffusive fluxes of $CH_4$ nor
$N_2O$ varied over the course of a 24-hour period.

For diffusive $CH_4$ flux, the overall trend was towards significantly higher wet season (51.1 ±
7.0 mg $CH_4$-C $m^{-2}$ $d^{-1}$) compared to dry season (27.3 ± 2.7 mg $CH_4$-C $m^{-2}$ $d^{-1}$) flux (data pooled
across all vegetation types; t-Test, $P < 0.001$, Table 4). However, when diffusive $CH_4$ flux was
disaggregated by vegetation type, very different seasonal trends emerged. For example, both
forested vegetation and mixed palm swamp showed significantly greater diffusive $CH_4$ flux
during the *dry season* with net fluxes of 47.2 ± 5.4 mg $CH_4$-C $m^{-2}$ $d^{-1}$ and 64.2 ± 12.1 mg $CH_4$-
C $m^{-2}$ $d^{-1}$, respectively (Fisher's LSD, $P < 0.05$, Table 3). In contrast, *wet season* flux was 7-16
times lower, with net fluxes of 6.7 ± 1.0 mg $CH_4$-C $m^{-2}$ $d^{-1}$ and 6.1 ± 1.3 mg $CH_4$-C $m^{-2}$ $d^{-1}$,
respectively (Fisher's LSD, $P < 0.05$, Table 3). In contrast, forested (short pole) vegetation and
*M. flexuosa* palm swamp showed seasonal trends consistent with the pooled data set; i.e.
significantly higher flux during the wet season (46.7 ± 8.4 and 60.4 ± 9.1 mg $CH_4$-C $m^{-2}$ $d^{-1}$,
respectively) compared to the dry season (28.3 ± 2.6 and 18.8 ± 2.6 mg $CH_4$-C $m^{-2}$ $d^{-1}$,
respectively) (Fisher's LSD, $P < 0.05$, Table 3).

Even though seasonal trends in $CH_4$ ebullition were not statistically significant, we will briefly
describe the overall patterns for the different vegetation types as they varied among
ecosystems (Table 2). Forested vegetation only showed evidence of ebullition during the dry
season, where ebullition-driven uptake was observed. For forested (short pole) vegetation,
net ebullition was generally greater during the wet season, while ebullition-driven uptake was
higher during the dry season. For *M. flexuosa* palm swamp, both net ebullition and ebullition-
driven uptake were greater during the wet season. Lastly, for mixed palm swamp, both net
ebullition and ebullition-driven uptake were greater during the dry season.

For the environmental variables, soil pH, DO, EC, water table depth, and soil temperature
varied significantly between seasons, whereas air temperature did not. Thus, for sake of
brevity, air temperature is not discussed further here. Mean soil pH was significantly lower
during the wet season (5.18 ± 0.03) than during the dry season (5.31 ± 0.04) (data pooled
across all vegetation types; t-Test, $P < 0.05$, Table 2). When disaggregated by vegetation type,
the overall trend was found to hold true for all vegetation types except forested (short pole)
vegetation, which displayed higher pH during the wet season compared to the dry season
(Table 2). A two-way ANOVA on Box-Cox transformed data using vegetation type, season and
their interaction as explanatory variables indicated that vegetation type was the best
predictor of pH, with season and vegetation type by season playing a lesser role ($F_{7,\ 1166}$ =
348.9, $P < 0.0001$).

For DO, the overall trend was towards significantly lower DO during the wet season (13.9 ±
1.0 %) compared to the dry season (19.3 ± 1.2 %) (data pooled across all vegetation types;
Wilcoxon test, $P < 0.0001$, Table 2). However, when the data were disaggregated by
vegetation type, we found that individual vegetation types showed distinct seasonal trends
from each other. Forested vegetation and mixed palm swamp were consistent with the
overall trend (i.e. lower wet season compared to dry season DO), whereas forested (short
pole) vegetation and *M. flexuosa* palm swamp displayed the reverse trend (i.e. higher *wet*
*season* compared to *dry season* DO) (Table 2). A two-way ANOVA on Box Cox transformed
data using vegetation type, season and their interaction as explanatory variables indicated
that vegetation type was the best predictor of DO, followed by a strong vegetation by season
interaction; season itself played a lesser role than either of the other two explanatory
variables ($F_{7,\ 1166}$ = 57.0, $P < 0.0001$).

For EC, the overall trend was towards lower EC in the wet season (49.4 ± 1.8 µs m$^{-2}$) compared
to the dry season (65.5 ± 2.2 µs m$^{-2}$) (data pooled across all vegetation types; Wilcoxon test,
$P < 0.05$, Table 2). When the data were disaggregated by vegetation type, this trend was
consistent for all the vegetation types except for forested vegetation, where differences
between wet and dry season were not statistically significant (Wilcoxon, $P > 0.05$, Table 2).

Water table depths varied significantly between seasons (data pooled across all vegetation
types; Wilcoxon test, $P < 0.0001$, Table 2). Mean water table level was significantly higher in
the wet (54.1 ± 2.7 cm) than the dry (1.3 ± 0.8 cm) season. When disaggregated by vegetation
type, the trend held true for individual vegetation types (Table 2). All vegetation types had
negative dry season water tables (i.e. below the soil surface) and positive wet season water
tables (i.e. water table above the soil surface), except for *M. flexuosa* palm swamp that had
positive water tables in both seasons. Two-way ANOVA on Box-Cox transformed data using
vegetation type, season and their interaction as explanatory variables indicated that all three
factors explained water table depth, but that season accounted for the largest proportion of
the variance in the model, followed by vegetation by season, and lastly by vegetation type ($F_{7,}$
$_{1157}$ = 440.1, $P < 0.0001$).

For soil temperature, the overall trend was towards slightly higher temperatures in the wet
season (25.6 ± 0.0 °C) compared to the dry season (25.1 ± 0.0 °C) (t-Test, $P < 0.0001$). Analysis
of the disaggregated data indicates this trend was consistent for individual vegetation types
(Table 2). Two-way ANOVA on Box-Cox transformed data using vegetation type, season and
their interaction as explanatory variables indicated that all three variables played a significant
role in modulating soil temperature, although season accounted for the largest proportion of
the variance whereas the other two factors accounted for a similar proportion of the variance
($F_{7, 1166}$ = 21.3, $P < 0.0001$).

**5.3 Relationships between gas fluxes and environmental variables**
To explore the relationships between environmental variables and diffusive gas fluxes, we
conducted an analysis of covariance (ANCOVA) on Box-Cox transformed gas flux data, using
vegetation type, season, vegetation by season, and environmental variables as explanatory
variables. We did not analyze trends between ebullition and environmental variables because
of the limitations in the sampling methodology and the limited number of observations.

For diffusive $CH_4$ flux, ANCOVA revealed that vegetation by season was the strongest
predictor of $CH_4$ flux, followed by a strong season effect ($F_{13,\ 917}$ = 9.2, $P<0.0001$). Other
significant drivers included soil temperature, water table depth, and a borderline-significant
effect of vegetation type ($P < 0.06$). However, it is important to note that each of these
environmental variables were only weakly correlated with $CH_4$ flux even if the relationships
were statistically significant; for example, when individual bivariate regressions were
calculated, the $r^2$ values were less than 0.01 for each plot (see Supplementary Online
Materials, Figures S1 and S2).

For diffusive $N_2O$ flux, ANCOVA indicated that the best predictors of flux rates were dissolved
oxygen and electrical conductivity ($F_{13,\ 1014}$= 2.2, $P < 0.0082$). As was the case for $CH_4$, when
the relationships between these environmental variables and $N_2O$ flux were explored using
individual bivariate regressions, $r^2$ values were found to be very low (e.g. less than $r^2 < 0.0007$)
or not statistically significant (see Supplementary Online Materials, Figures S3 and S4).


**6. DISCUSSION**
**6.1 Large and asynchronous $CH_4$ fluxes from peatlands in the Pastaza-Marañón foreland**
**basin**
The ecosystems sampled in this study were strong atmospheric sources of $CH_4$. Diffusive $CH_4$
flux, averaged across all vegetation types, was 36.1 ± 3.1 mg $CH_4$-C $m^{-2} d^{-1}$, spanning a range
from -100 to 1,510 mg $CH_4$-C $m^{-2} d^{-1}$. This mean falls within the range of other diffusive fluxes
observed in Indonesian peatlands (3.7-87.8 mg $CH_4$-C $m^{-2} d^{-1}$) (Couwenberg et al., 2010) and
other Amazonian wetlands (7.1-390.0 mg $CH_4$-C $m^{-2} d^{-1}$) (Bartlett et al., 1990;Bartlett et al.,
1988;Devol et al., 1990;Devol et al., 1988). Although the ebullition data must be treated with
caution because of the sampling methodology (see below), we observed a mean net ebullition
flux of 973.3 ± 161.4 mg $CH_4$-C $m^{-2} d^{-1}$, spanning a range of 27 to 8,082 mg $CH_4$-C $m^{-2} d^{-}1$.
While data on ebullition from Amazonian wetlands are sparse, these values are broadly in-
line with riverine and lake ecosystems sampled elsewhere (Bastviken et al., 2010;Smith et al.,
2000;Sawakuchi et al., 2014). Ebullition-driven $CH_4$ uptake is not a commonly reported
phenomena in other peatland studies because it is likely an artefact of chamber sampling
methods; as a consequence, we do not discuss these data further here. To summarize, these
data on diffusive $CH_4$ flux and ebullition suggest that peatlands in the Pastaza-Marañón
foreland basin are strong contributors to the regional atmospheric budget of $CH_4$, given that
the four vegetation types sampled here represent the dominant cover types in the PMFB
(Draper et al., 2014;Householder et al., 2012;Kelly et al., 2014;Lahteenoja and Page, 2011)

The overall trend in the diffusive flux data was towards greater temporal (i.e. seasonal)
variability in diffusive $CH_4$ flux rather than strong spatial (i.e. inter-site) variability. For the
pooled dataset, diffusive $CH_4$ emissions were significantly greater during the wet season than
the dry season, with emissions falling by approximately half from one season to the other (i.e.
51.1 ± 7.0 to 27.3 ± 2.7 mg $CH_4$-C $m^{-2}\,d^{-1}$). This is in contrast to the data on diffusive $CH_4$ flux
among study sites, where statistical analyses indicate that there was a weak effect of
vegetation type on $CH_4$ flux, that was only on the edge of statistical significance (i.e. ANCOVA;
$P < 0.06$ for the vegetation effect term). For the ebullition data, while there was no significant
difference among vegetation types nor between seasons, it is interesting to note that
ebullition was more common for the two vegetation types – Mixed Palm Swamp and *M.*
*flexuosa* palm swamp – that showed the highest rates of diffusive $CH_4$ flux (Figure 2, Table 2).
In contrast, forested (short pole) and forested vegetation, which showed the lowest rates of
diffusive $CH_4$ flux, also showed the lowest occurrence of ebullition (Figure 2, Table 2). This is
broadly consistent with the notion that Mixed Palm Swamp and *M. flexuosa* palm swamp may
produce more $CH_4$ or possess lower gross $CH_4$ oxidation rates than the other vegetation types.

On face value, these data on diffusive $CH_4$ flux suggest two findings: first, the relatively weak
effect of vegetation type on diffusive $CH_4$ flux implies that patterns of $CH_4$ cycling are broadly
similar among study sites. Second, the strong *overall* seasonal pattern suggests that – on the
whole – these systems conform to our normative expectations of how peatlands function
with respect to seasonal variations in hydrology and redox potential; i.e. enhanced $CH_4$
emissions during a more anoxic wet season (i.e. when water tables rise), and reduced $CH_4$
emissions during a more oxic dry season (i.e. when water tables fall). However, closer
inspection of the data reveals that different vegetation types showed contrasting seasonal
emission patterns (Table 3), challenging our basic assumptions about how these ecosystems
function. For example, while forested (short pole) vegetation and *M. flexuosa* palm swamp
conformed to expected seasonal trends for methanogenic wetlands (i.e. higher wet season
compared to dry season emissions), forested vegetation and mixed palm swamp showed the
opposite pattern, with significantly greater $CH_4$ emissions during the dry season. The
disaggregated data thus imply that the process-based controls on $CH_4$ fluxes may vary
significantly among these different ecosystems, rather than being similar, leading to a
divergence in seasonal flux patterns.

What may explain this pattern of seasonal divergence in $CH_4$ flux? One explanation is that $CH_4$
emissions from forested vegetation and mixed palm swamp, compared to the other two
ecosystems, may be more strongly transport-limited during the wet season than the dry
season. This interpretation is supported by the field data; forested vegetation and mixed palm
swamp had the highest wet season water table levels, measuring 110.8 ± 9.3 and 183.7 ± 1.7
cm, respectively (Table 2). In contrast, water table levels for forested (short pole) vegetation
and *M. flexuosa* palm swamp in the wet season were 3-7 times lower, measuring only 26.9 ±
0.5 and 37.2 ± 1.7 cm, respectively (Table 2). Moreover, a scatter plot of diffusive $CH_4$ flux
against water table depth shows a peak in diffusive $CH_4$ emissions when water tables are
between 30 to 40 cm above the surface, after which $CH_4$ emissions decline precipitously
(Supplementary Online Materials Figure S2). Thus, the greater depth of overlying water in
forested vegetation and mixed palm swamp may have exerted a much greater physical
constraint on gas transport compared to the other two ecosystems. This interpretation is
broadly consistent with studies from other ecosystems, which indicate that high or positive
water tables may suppress $CH_4$ emissions from wetlands above a system-specific threshold
(Couwenberg et al., 2010;Couwenberg et al., 2011).

However, transport limitation alone does not fully explain the difference in dry season $CH_4$
emissions among vegetation types. Forested vegetation and mixed palm swamp showed
substantially higher dry season $CH_4$ emissions (47.2 ± 5.4 and 85.5 ± 26.4 mg $CH_4$-C $m^{-2} d^{-1}$,
respectively) compared to forested (short pole) vegetation and *M. flexuosa* palm swamp (9.6
± 2.6 and 25.5 ± 2.9 mg $CH_4$-C $m^{-2} d^{-1}$, respectively), pointing to underlying differences in $CH_4$
production and oxidation among these ecosystems. One possibility is that dry season
methanogenesis in forested vegetation and mixed palm swamp was greater than in the other
two ecosystems, potentially driven by higher rates of C flow (Whiting and Chanton, 1993).
This is plausible given that forested vegetation and mixed palm swamp tend to occur in more
nutrient-rich parts of the Pastaza-Marañón foreland basin, whereas forested (short pole)
vegetation and *M. flexuosa* palm swamp tend to dominate in more nutrient-poor areas
(Lahteenoja et al., 2009a), leading to potential differences in rates of plant productivity and
belowground C flow. Moreover, it is possible that the nutrient-rich vegetation may be able to
utilize the higher concentration of nutrients, deposited during the flood pulse, during the
Amazonian dry season (Morton et al., 2014;Saleska et al., 2016), with implications for overall
ecosystem C throughput and $CH_4$ emissions. Of course, this interpretation does not preclude
other explanations, such as differences in $CH_4$ transport rates among ecosystems (e.g. due to
plant-facilitated transport or ebullition) (Panagala et al., 2013), or varying rates of $CH_4$
oxidation (Teh et al., 2005). However, these other possibilities cannot be explored further
without recourse to more detailed process-level experiments. Forthcoming studies on the
regulation of GHG fluxes at finer spatial scales (e.g. investigation of environmental gradients
within individual study sites) or detailed diurnal studies of GHG exchange (Murphy *et al.*, in
prep.) will further deepen our understanding of the process controls on soil GHG flux from
these peatlands, and shed light on these questions.

Finally, while the trends described here are intriguing, it is important to acknowledge some
of the potential limitations of our data. First, given the uneven sampling pattern, it is possible
that the values reported here do not fully represent the entire range of diffusive flux rates,
especially for the more sparsely sampled habitats. However, given the large and statistically
significant differences in $CH_4$ emissions between seasons, it is likely that the main trends that
we have identified will hold true with more spatially-extensive sampling. Second, the data are
a conservative underestimate of $CH_4$ emissions, because the low frequency, static chamber
sampling approach that we utilized was unable to fully capture erratic ebullition events
representatively (McClain et al., 2003). Although we attempted to quantify $CH_4$ ebullition
within our static flux chambers, the sampling approach that we utilized was not the best-
suited for representatively quantifying ebullition. Given the erratic or stochastic nature of
ebullition, automated chamber measurements or an inverted "flux funnel" approach  would
have provided better estimates of ebullition (Strack et al., 2005). However, we lacked the
resources to apply these techniques here. We also did not measure $CH_4$ emissions from the
stems of woody plants, even though woody plants have been recently identified as an
important point of atmospheric egress (Pangala et al., 2013). We did not have enough data
on floristic composition or individual plant identities within our plots to develop a sampling
design that would adequately represent plant-mediated fluxes from our study sites, nor the
resources to implement a separate study of stem fluxes. Third and last, our data probably
underestimate net $CH_4$ fluxes for the PMFB because we chose to include fluxes with strong
negative values (i.e. more than -10 mg $CH_4$-C $m^{-2}d^{-1}$) in our calculation of mean diffusive flux
rates. These observations are more negative than other values typically reported elsewhere
in the tropical wetland literature (Bartlett et al., 1990;Bartlett et al., 1988;Devol et al.,
1990;Devol et al., 1988;Couwenberg et al., 2010). However, they represent only a small
proportion of our dataset (i.e. 7 %, or only 68 out of 980 measurements), and inspection of
our field notes and the data itself did not produce convincing reasons to exclude these
observations (e.g. we found no evidence of irregularities during field sampling, and any
chambers that showed statistically insignificant changes in concentration over time were
removed during our quality control procedures). While headspace concentrations for these
measurements were often elevated above mean tropospheric levels (>2 ppm), this in itself is
not unusual in reducing environments that contain strong local sources of $CH_4$ (Baldocchi et
al., 2012). We did not see this as a reason to omit these values as local concentrations of $CH_4$
are likely to vary naturally in methanogenic forest environments due to poor mixing in the
understory and episodic ebullition events. Importantly, exclusion of these data did not alter
the overall statistical trends reported above, and only produced slightly higher estimates of
diffusive $CH_4$ flux (41.6 ± 3.2 mg $CH_4$-C $m^{-2}d^{-1}$ versus 36.1 ± 3.1 mg $CH_4$-C $m^{-2}d^{-1}$).

**6.2 Western Amazonian peatlands as weak atmospheric sources of nitrous oxide**
The ecosystems sampled in this study were negligible atmospheric sources of $N_2O$, emitting
only 0.70 ± 0.34 µg $N_2O$-N $m^{-2}d^{-1}$, suggesting that peatlands in the Pastaza-Marañón foreland
basin make little or no contribution to regional atmospheric budgets of $N_2O$. This is consistent
with $N_2O$ flux measurements from other forested tropical peatlands, where $N_2O$ emissions
were also found to be relatively low (Inubushi et al., 2003; Couwenberg et al., 2010). No
statistically significant differences in $N_2O$ flux were observed among study sites or between
seasons, suggesting that these different peatlands may have similar patterns of $N_2O$ cycling.
Interestingly, differences in $N_2O$ fluxes were not associated with the nutrient status of the
peatland; i.e. more nutrient-rich ecosystems, such as forested vegetation and mixed palm
swamp, did not show higher $N_2O$ fluxes than their nutrient-poor counterparts, such as
forested (short pole) vegetation and *M. flexuosa* palm swamp. This may imply that N
availability, one of the principal drivers of nitrification, denitrification, and $N_2O$ production
(Groffman et al., 2009;Werner et al., 2007), may not be greater in nutrient-rich versus
nutrient-poor ecosystems in this part of the Western Amazon. Alternatively, it is possible that
even though N availability and N fluxes may differ between nutrient-rich and nutrient-poor
systems, $N_2O$ yield may also vary such that net $N_2O$ emissions are not significantly different
among study sites (Teh et al., 2014).

One potential source of concern are the negative $N_2O$ fluxes that we documented here. While
some investigators have attributed negative fluxes to instrumental error (Cowan et al.,
2014;Chapuis-Lardy et al., 2007), others have demonstrated that $N_2O$ consumption –
particularly in wetland soils – is not an experimental artifact, but occurs due to the complex
effects of redox, organic carbon content, nitrate availability, and soil transport processes on
denitrification (Ye and Horwath, 2016;Yang et al., 2011;Wen et al., 2016;Schlesinger,
2013;Teh et al., 2014;Chapuis-Lardy et al., 2007). Given the low redox potential and high
carbon content of these soils, it is plausible that microbial $N_2O$ consumption is occurring,
because these types of conditions have been found to be conducive for $N_2O$ uptake elsewhere
(Ye and Horwath, 2016;Teh et al., 2014;Yang et al., 2011).


**7. CONCLUSIONS**
Our data suggest that peatlands in the Pastaza-Marañón foreland basin are strong sources of
atmospheric $CH_4$ at a regional scale, and need to be better accounted for in $CH_4$ emissions
inventories for the Amazon basin as a whole. In contrast, $N_2O$ fluxes were negligible,
suggesting that these ecosystems are weak regional sources at best. Divergent or
asynchronous seasonal emissions pattern for $CH_4$ among different vegetation types was
intriguing, and challenges our underlying expectations of how tropical peatlands function.
These data highlight the need for greater wet season sampling, particularly from ecosystems
near river margins that may experience very high water tables (i.e. >40 cm). Moreover, these
data also emphasize the need for more spatially-extensive sampling across both the Pastaza-
Marañón foreland basin and the wider Amazon region as a whole, in order to establish if these
asynchronous seasonal emission patterns are commonplace or specific to peatlands in the
PMFB region. If $CH_4$ emission patterns for different peatlands in the Amazon are in fact
asynchronous and decoupled from rainfall seasonality, then this may partially explain some
of the heterogeneity in $CH_4$ source and sinks observed at the basin-wide scale (Wilson et al.,

698 2016).



**8. AUTHOR CONTRIBUTION**

YAT secured the funding for this research, assisted in the planning and design of the experiment, and took the principal role in the analysis of the data and preparation of the manuscript. WAM planned and designed the experiment, collected the field data, analyzed the samples, and took a secondary role in data preparation, data analysis, and manuscript preparation. JCB, AB, and SEP supported the planning and design of the experiment, and provided substantive input into the writing of the manuscript. AB in particular took a lead role in developing the maps of our study sites in the PMFB.

**9. ACKNOWLEDGEMENTS**

The authors would like to acknowledge the UK Natural Environment Research Council for funding this research (NERC award number NE/I015469). We would like to thank MINAG and the Ministerio de Turismo in Iquitos for permits to conduct this research, the Instituto de Investigaciones de la Amazonía Peruana (IIAP) for logistical support, Peruvian rainforest villagers for their warm welcome and acceptance, Hugo Vasquez, Pierro Vasquez, Gian Carlo Padilla Tenazoa and Yully Rojas Reátegui for fieldwork assistance, Dr Outi Lahteenoja and Dr Ethan Householder for fieldwork planning, and Dr Paul Beaver of Amazonia Expeditions for lodging and logistical support. Our gratitude also goes to Alex Cumming for fieldwork support and laboratory assistance, Bill Hickin, Gemma Black, Adam Cox, Charlotte Langley, Kerry Allen, and Lisa Barber of the University of Leicester for all of their continued support. Thanks are also owed to Graham Hambley (St Andrews), Angus Calder (St Andrews), Viktoria Oliver

(Aberdeen), Torsten Diem (Aberdeen), Tom Kelly (Leeds), and Freddie Draper Leeds) for their
help in the laboratory and with fieldwork planning. TD, VO, and two anonymous referees
provided very helpful and constructive comments on earlier drafts of this manuscript. This
publication is a contribution from the Scottish Alliance for Geoscience, Environment and
Society (http://www.sages.ac.uk) and the UK Tropical Peatland Working Group
(https://tropicalpeat.wordpress.com).

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

**Table 1.** Site characteristics including field site location, nutrient status, plot and flux chamber
replication

| Vegetation type | Site name | Nutrient status* | Latitude (S) | Longitude (W) | Plots | Flux chambers |
|---|---|---|---|---|---|---|
| Forested | Buena Vista | Rich | 4°14'45.60"S | 73°12'0.20"W | 21 | 105 |
| Forested (short pole) | San Jorge (centre) | Poor | 4°03'35.95"S | 73°12'01.13"W | 6 | 28 |
| Forested (short pole) | Miraflores | Poor | 4°28'16.59"S | 74° 4'39.95"W | 41 | 204 |
| M. flexuosa Palm Swamp | Quistococha | Intermediate | 3°49'57.61"S | 73°12'01.13"W | 135 | 668 |
| M. flexuosa Palm Swamp | San Jorge (edge) | Intermediate | 4°03'18.83"S | 73°10'16.80"W | 18 | 86 |
| Mixed palm swamp | Charo | Rich | 4°16'21.80"S | 73°15'27.80"W | 18 | 90 |

*After Householder et al. 2012, Lahteenoja et al. 2009a, and Lahteenoja et al. 2009b

**Table 2.** Proportion of observations for each vegetation type that showed evidence of
ebullition, mean rates of ebullition and ebullition-driven CH$_4$ uptake. Values represent
means and standard errors.

| Vegetation Type | Percentage of observations | Net Ebullition (mg CH$_4$-C m$^{-2}$ d$^{-1}$) | | Ebullition-driven uptake (mg CH$_4$-C m$^{-2}$ d$^{-1}$) | |
|---|---|---|---|---|---|
| | (%) | Wet Season | Dry Season | Wet Season | Dry Season |
| Forested | 10.5 | 0 | 0 | 0 | -136.4 ± 0.1 |
| Forested (short pole) | 6.9 | 994.6 ± 293.2 | 512.5 ± 153.0 | -95.8 ± 0.0 | -245.5 ± 48.9 |
| *M. flexuosa* Palm Swamp | 16.7 | 1192.0 ± 305.7 | 994.3 ± 237.3 | -869.4 ± 264.8 | -401.4 ± 59.9 |
| Mixed Palm Swamp | 12.2 | 0 | 733.6 ± 313.1 | 0 | -464.4 ± 565.9 |


**Table 3.** Environmental variables for each vegetation type for the wet and dry season.
Values reported here are means and standard errors. Lower case letters indicate significant
differences among vegetation types within the wet or dry season (Fisher's LSD, $P < 0.05$).

| Vegetation Type | Peat Temperature (°C) | | Air Temperature (°C) | | Conductivity ($\mu S\ m^{-2}$) | | Dissolved Oxygen (%) | | Water Table Level (cm) | | pH | |
|---|---|---|---|---|---|---|---|---|---|---|---|---|
| | Wet Season | Dry Season | Wet Season | Dry Season | Wet Season | Dry Season | Wet Season | Dry Season | Wet Season | Dry Season | Wet Season | Dry Season |
| Forested | 26.1 ± 0.1a | 24.7 ± 0.0a | 28.8 ± 0.7a | 26.4 ± 0.3a | 79.0 ± 5.9a | 75.9 ± 5.7a | 0.2 ± 0.1a | 18.9 ± 4.4a | 110.8 ± 9.3a | -13.2 ± 0.7a | 5.88 ± 0.15a | 6.31 ± 0.04a |
| Forested (short pole) | 25.2 ± 0.0b | 24.8 ± 0.1a | 27.6 ± 0.1b | 27.5 ± 0.1b | 21.0 ± 0.0b | 48.5 ± 4.8b | 4.4 ± 0.0a | 33.1 ± 2.6b | 26.9 ± 0.5b | -4.7 ± 0.4b | 4.88 ± 0.01b | 3.8 ± 0.03b |
| M. flexuosa Palm Swamp | 25.6 ± 0.6c | 25.3 ± 0.1b | 26.3 ± 0.1c | 26.4 ± 0.1a | 45.9 ± 2.1c | 51.9 ± 1.8b | 19.4 ± 1.3b | 17.3 ± 1.5a | 37.2 ± 1.7c | 6.1 ± 1.3c | 5.04 ± 0.03c | 5.49 ± 0.03c |
| Mixed Palm Swamp | 26.0 ± 0.0a | 25.0 ± 0.1ab | 26.1 ± 0.1c | 28.2 ± 0.3b | 100.0 ± 0.2d | 206.4 ± 4.2c | 0.0 ± 0.0a | 0.0 ± 0.0c | 183.7 ± 1.7d | -2.4 ± 0.3b | 6.1 ± 0.03a | 6.82 ± 0.02d |


**Table 4.** Trace gas fluxes for each vegetation type for the wet and dry season. Values reported
here are means and standard errors. Upper case letters indicate significant differences in gas
flux between seasons with a vegetation type, while lower case letters indicate significant
differences among vegetation types within a season (Fisher's LSD, $P < 0.05$).

| Vegetation Type | Methane Flux (mg $CH_4$-C $m^{-2}$ $d^{-1}$) | | Nitrous Oxide Flux ($\mu$g $N_2$O-N $m^{-2}$ $d^{-1}$) | |
|---|---|---|---|---|
| | Wet Season | Dry Season | Wet Season | Dry Season |
| Forested | 6.7 ± 1.0Aa | 47.2 ± 5.4Ba | 2.54 ± 1.48 | -1.16 ± 1.20 |
| Forested (short pole) | 60.4 ± 9.1Ab | 18.8 ± 2.6Bb | 1.16 ± 0.54 | -0.42 ± 0.90 |
| M. flexuosa Palm Swamp | 46.7 ± 8.4Ac | 28.3 ± 2.6Bc | 1.14 ± 0.35 | 0.92 ± 0.61 |
| Mixed Palm Swamp | 6.1 ± 1.3Aa | 64.2 ± 12.1Ba | 1.45 ± 0.79 | -0.80 ± 0.79 |


 **Figure Captions**

**Figure 1.** Map of the study region and field sites. The colour scale to the right of the map
denotes elevation in meters above sea level (m a.s.l.). Tan and brown tones indicate areas in
which peatlands are found; however, not all of these areas are peatland-dominated.

**Figure 2.** Net diffusive **(a)** methane ($CH_4$) and **(b)** nitrous oxide **(**$N_2O$) fluxes by vegetation type.
Error bars denote standard errors.
**Figure 1**

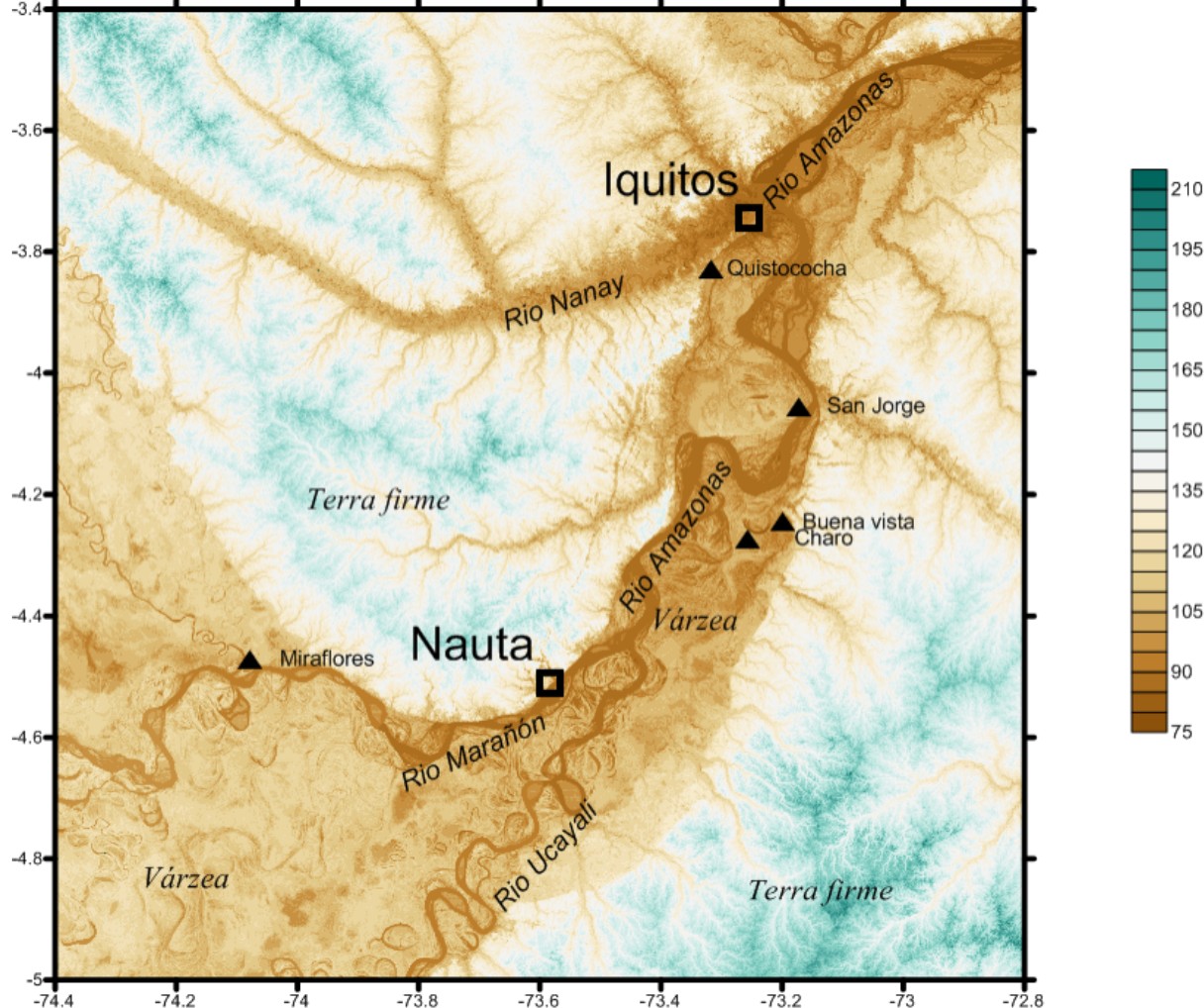

**Figure 2**

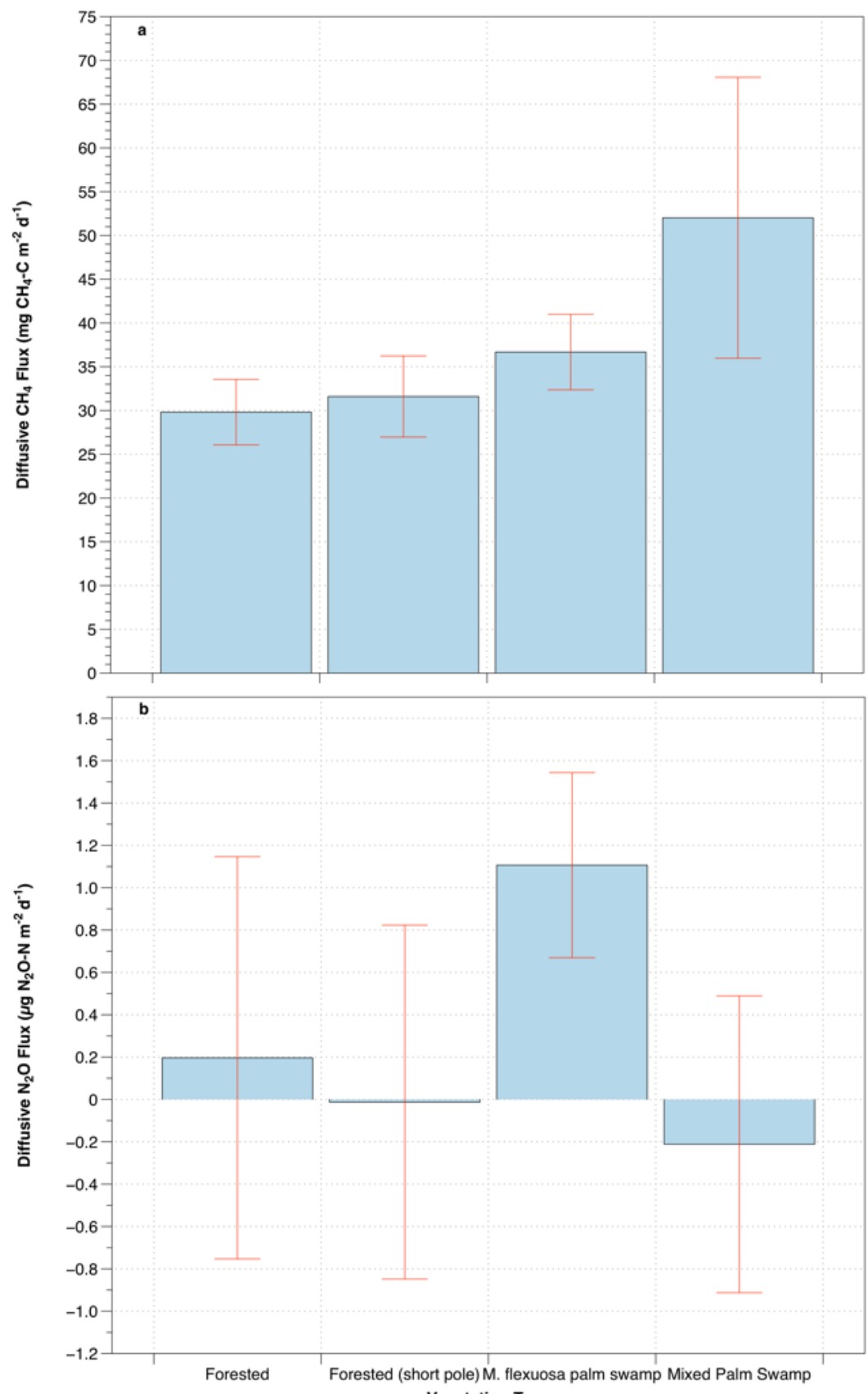
