# Peer review of "1. TITLE PAGE Seasonal variability in methane and nitrous oxide fluxes from tropical peatlands in the Western Amazon basin"

_Biogeosciences, 2017_

## Referee Comment (RC1) · Anonymous Referee #1 · 7 Mar 2017

Although the number of studies on trace gas fluxes from lowland tropical peatlands are steadily increasing, measurements are completely absent from some important large peatland areas. The peats in the foreland basin of Pastaza Maranon (PMFB) in Peru is an example of an important tropical peatland area from which we have no empirical studies on CH4 and N2O fluxes. As such, this study is timely and novel. The authors conducted chamber based CH4 and N2O fluxes stratified according to the four dominant vegetation types ranging from nutrient rich floodplains to nutrient poor bogs. While the spatial resolution of the sampling is good (see Table 1) the temporal resolution is restricted to four time sampling over a period of two years (sampling twice during the wet season and twice during a dry season). However, given the difficult

accessibility and the fact that this is the first report on fluxes from this important peat area this is acceptable for this initial study. Furthermore, we do learn important controls since supporting variables were measured as well. Nevertheless if have several point that should be addressed before the manuscript can be accepted for publication.

-In your introduction, I miss a section on the known controls of CH4 and N2O fluxes in peatlands. I think it is important to include this, since it is otherwise unclear why you measured the environmental variables that you did. Are there some potentially important factors that you did not measure and if so, why was that the case?

-In l.142 and l.147 you mention soil Ca, it remains a mystery in which form this Ca occurs. Please provide more details, is Ca an important environmental control on CH4 or N2O fluxes?

-You describe your chambers as floating static chambers (l. 179). You furthermore write that chamber were placed on the soil surface from a distance of no closer than 2m to reduce the risk of ebullition (l. 186). For me it is hard to believe that ebullition was completely excluded and I also cannot understand how you can place these chambers at a distance of 2 m? Nobody has arms that long (at least I don't), so how was this done in detail? And, especially, how did you take the gas samples from your chambers using syringes without causing ebullition?

-Ebullition is relatively easy to detect e.g. if you start with very high concentrations or if you detect abrupt increases in CH4 concentration. Can you give us information on how you dealt which such data and how many of your chamber measurements were potentially affected by ebullition? Is there a correlation of flux strength and the potential occurrence of ebullition? You also mention in your discussion that you measured occasionally high CH4 uptake values (l. 477). Is it possible that this was related to high CH4 concentrations at the beginning of chamber closure, potentially caused by ebullition? Did you find a correlation between initial CH4 concentration and calculated CH4 uptake values? I was also wondering whether the contrasting seasonal emission

patterns that you discuss (l. 417 and further) could be related to ebullition. As you see, I think you don't give use sufficient information about the potential occurrence of ebullition and you should clearly improve this.

-I have no problem with the fact that you measured negative N2O fluxes, since we see more and more evidence that globally this is quite an important process. However, some of the negative fluxes seem to be quite high in your figure 2. How do your N2O uptake rates compare to other published values and is it possible that this may also somehow related to ebullition? Also, here I would be interested if your negative values correlate with the initially measured concentration in your chambers. If ebullition played a role for negative Ch4 and N2O fluxes, you may expect that both strong CH4 uptake and N2O uptake would occur in the same chamber measurement. Did you check this?

-You probably measured your fluxes at different time of the day. Did you find any diurnal pattern in GHG fluxes? If yes, what could explain these observed differences and how would this affect your estimate of emission strength?

-The potential role of CH4 oxidation is remarkable absent from your discussion. Why is that the case? Do you think this is not important for the total measured CH4 fluxes?

-The version of the map in Fig. 1 that I saw did not have a very good quality. Can you provide a map where the four vegetation types that you used for stratification are included? The impression of Fig. 1 is that the total peat area is larger than what can be seen in the map. Can you adjust Fig. 1 in such a way that we see all of (or most of) the PMFB peatland area?

---

## Referee Comment (RC2) · Anonymous Referee #2 · 14 Mar 2017

General comments

The investigations by Teh et al. on CH4 and N2O emissions from tropical peatlands are recently of major interest. Particularly for the study region in the Amazon basin, knowledge on magnitude, pattern and controls of greenhouse gas fluxes is scarce. As this region is a potentially huge source of CH4, it is important to close this knowledge gap. This study could contribute to this process. The findings of large CH4 and negligible N2O emissions might have been anticipated while the asynchronous CH4 fluxes with higher fluxes during the dry season for two of the four investigated vegetation types might not. Potential explanations for this result are discussed. However, there is plenty of literature on mechanisms and controls of peatland CH4 and N2O fluxes but

appropriate references are missing in the introduction as well as in the discussion part. For example, it has been reported that CH4 fluxes do not increase or even decline when sites are inundating and that highest emissions occur for water levels close to the surface (e.g. Couwenberg et al., 2011, Hydrobiologia 674, 67-89).

Furthermore, I strongly recommend to revise the abstract and the presentation of results. The abstract mainly lists the results but doesn't tell anything about motivation, objective and main conclusions of the study. The two figures are of poor quality and Fig. 2 is not very helpful for interpretation of results due to the huge amounts of outliers. I would recommend to rather show columns with standard errors or maxima/minima. And why was the data shown in the figure grouped for vegetation type but not for different seasons? Also, figures that visualize the relationships between GHG fluxes and the measured environmental variables would be interesting. Although the relationships might be very weak, this would give the reader a better idea of the dataset.

Specific comments

P4, L60: Peatlands are not necessarily peat-forming. In contrast, many peatlands have been drained for utilization purposes which turns them into significant C sources and in regions like Central Europe, only a few percent of peatlands are still in a natural condition and thus peat-forming. Please be more specific about the state of peatlands in the study region.

P4, L61-65: Are you talking about peatlands in general or about peatlands in the Amazon basin? If you mean peatlands in general, your statements are not correct as there are several studies on peatland GHG emissions but most of them were conducted in the boreal or temperate zone.

P5, L90: I wouldn't classify a peat depth of 3.9 m as shallow. Following the international definition, peatlands are defined by a minimum of 20 cm of peat deposit, which could be classified as shallow. It seems that peat deposits in the study region are all very deep.

P7, L127-128: You do not mention the months of September and October. Are they neither wet season nor dry season?

P7, L135: "Pure peat" is not a soil classification! Please use World reference base (2015) to classify your sites. Qualifiers should be used to characterize the soils more precisely.

P7, L136: The pH values given are not in line with Table 2. Are these values from preliminary measurements?

P8, L150: The number of plots remains unclear to me. You mention 229 plots but the numbers below summarize to 148 plots and in Table 1 you list 161 plots. Please clarify.

P9, L177: Please clarify the procedure of the measurement campaigns. How long were these campaigns, did you sample each plot only once, several times per day or also on different days?

P9, L178: What about the wet season in 2013? Why didn't you measure the fluxes during that period?

P9, L186: How is it possible to place the chamber in a distance of 2 m? I cannot imagine how this practically works. And what about the sampling procedure? You have to get quite close to the chambers for that. Please clarify.

P10, L199: Does it mean that fluxes were calculated from linear or non-linear regressions depending on the individual concentration trend against time? It is important to clarify this as linear regression can lead to substantial underestimation of fluxes as a consequence of decreasing concentration gradients over time. And which quality criteria have been used to ensure the reliability of computed fluxes?

P13, L268: The paragraphs for the results of different variables always have the same wording, which gives a quite uninspired impression.

P14, L281: I don't find it very meaningful to do statistics on measurements of air temperature. Also, you would have to compare exactly the same periods, otherwise the results could be misleading.

P16, L325: Several different statistical tests were applied but not mentioned in the statistical analyses section.

P19, L398: It has to be considered that conclusions on that can only be drawn when annual CH4 budgets can be estimated from regular or automatic measurements in high temporal resolution.

P20, L 416: The water tables of the studied sites do in my opinion not allow the definition of oxic conditions as the water tables reported were quite high even in the dry season. Particularly non-degraded peat has a high water holding capacity, thus also when the water table drops below the soil surface, water filled pore space remains high in the top soil, still preventing CH4 oxidation.

P21, L 438: No references are given in this section. The weak relationship is probably a result of the overall high water levels as the general assumption that CH4 emissions increase with water level is based on measurements from sites with huge drainage gradients. Generally, CH4 emissions increase exponentially when the water level passes a threshold of roughly 20 – 30 cm below ground. For water levels close to or above the surface, CH4 fluxes are often extremely variable. Some references would be very helpful here for the interpretation of your results. Also, methodological issues should be discussed as CH4 emissions mainly occur in terms of erratic ebullition when water tables are above the soil surface. This might be difficult to detect with small chambers during a short period of enclosure.

P23, L479: Where negative CH4 fluxes also measured for water tables above ground? This would be rather unreliable in my opinion as one would not expect CH4 uptake in water saturated soil or even open water. Also, I assume that there should be more recent literature on CH4 exchange in tropical peatlands.

P24, L505-508: It is for me very unlikely that the different ecosystems do not differ in N availability. This conclusion cannot be drawn from equally low N2O emissions as there are probably other reasons for negligible N2O fluxes also in the nutrient-rich ecosystems. For example, there might be a higher N uptake by productive plant species at the nutrient-rich sites, competing with the potentially N2O producing microbes or N compounds are completely reduced to N2 during denitrification because of permanently anoxic conditions.

Technical corrections

P2, L9: The numbering of sections starts with 2.

P5, L99: Replace "positive water tables" by "high water tables" or "water tables above ground".

P10, L197: I assume that the instrumental precision was > 95 % or the instrumental noise was < 5 %.

P10, L210: In which height was the air temperature measured?

P10, L211: Please add manufacturer.

P13, L270: add "different" after "significantly".

P14, L282: "Soil temperature" has to be replaced by "air temperature".

P15, L314: Add "during" before "the dry season".

P18, L364: Plural: relationships

P18, L372: Results should not be interpreted in the "Results" section.

P18, L382: Add "electrical" before "conductivity".

P18, L383: Why do you mention CO2 here? No results on CO2 were shown.

P19, L394: Please round up to 1510.

P25, L526: Replace "these data" by "our data".
* * *

---

## Author Comment (AC1) · 20 Mar 2017

1. Although the number of studies on trace gas fluxes from lowland tropical peatlands are steadily increasing, measurements are completely absent from some important large peatland areas. The peats in the foreland basin of Pastaza Maranon (PMFB) in Peru is an example of an important tropical peatland area from which we have no empirical studies on CH4 and N2O fluxes. As such, this study is timely and novel. The authors conducted chamber based CH4 and N2O fluxes stratified according to the four dominant vegetation types ranging from nutrient rich floodplains to nutrient poor bogs. While the spatial resolution of the sampling is good (see Table 1) the temporal resolution is restricted to four time sampling over a period of two years (sampling twice

during the wet season and twice during a dry season). However, given the difficult accessibility and the fact that this is the first report on fluxes from this important peat area this is acceptable for this initial study. Furthermore, we do learn important controls since supporting variables were measured as well. Nevertheless if have several point that should be addressed before the manuscript can be accepted for publication.

Authors' response: The authors would like to thank the referee for his/her thoughtful and insightful comments on our manuscript, and welcome the opportunity to improve the manuscript for the wider readership of Biogeosciences.

2. In your introduction, I miss a section on the known controls of CH4 and N2O fluxes in peatlands. I think it is important to include this, since it is otherwise unclear why you measured the environmental variables that you did. Are there some potentially important factors that you did not measure and if so, why was that the case?

Authors' response: This was an oversight on our part. In order to keep the paper succinct, we omitted a more detailed review of the literature on the factors regulating CH4 and N2O fluxes. We will provide a more through description of the process-based on controls on CH4 and N2O flux in the revised version of the manuscript.

With respect to potentially important factors that we did not measure; we did not quantify CH4 fluxes from woody plants nor did we specifically seek to quantify ebullition. While plant-mediated fluxes are believed to be important in tropical wetland ecosystems (Pangala et al., 2013), we did not have enough data on floristic composition or individual plant identities within our plots to come-up with a sampling design that would adequately represent plant-mediated fluxes from our study sites. Likewise, ebullition is often characterized by high spatial and temporal variability. In order to develop a representative, spatially-stratified sampling design to quantify ebullition, using techniques such as the inverted "flux funnel" approach (Strack et al., 2005), more detailed information on spatial patterns in net CH4 flux would be required; information that we did not have prior to the collection of the data presented here. As a consequence, we chose

to omit specific measurements of ebullition from this study, with a wider view towards collecting these data at a later date.

For N2O, even though inorganic N is thought to be one of the major drivers of N2O flux, we did not collect data on inorganic N (NH4+, NO3-), because the relatively modest budget for this project did not accommodate costs for inorganic N analysis.

3. In l.142 and l.147 you mention soil Ca, it remains a mystery in which form this Ca occurs. Please provide more details, is Ca an important environmental control on CH4 or N2O fluxes?

Authors' response: The Ca in these systems occurs as dissolved inorganic Ca2+ associated with the soil or peat exchange complex, or Ca found in secondary minerals (Lahteenoja et al., 2009). As far as we are aware, Ca has no direct effect on CH4 or N2O fluxes, although Ca may indirectly influence trace gas exchange by influencing plant productivity and organic matter decay. We will revise the text to acknowledge this and remove any ambiguity from the manuscript.

4. You describe your chambers as floating static chambers (l. 179). You furthermore write that chamber were placed on the soil surface from a distance of no closer than 2m to reduce the risk of ebullition (l. 186). For me it is hard to believe that ebullition was completely excluded and I also cannot understand how you can place these chambers at a distance of 2 m? Nobody has arms that long (at least I don't), so how was this done in detail? And, especially, how did you take the gas samples from your chambers using syringes without causing ebullition?

Authors' response: The referee is correct that we were unable to entirely exclude ebullition from our dataset. We did in fact find evidence of ebullition, with 164 of the 1181 chamber observations (13.9 %) showing signs of ebullition (e.g. abrupt, non-linear changes in headspace concentrations). Of these 164 observations, 83 (7.0 %) showed net CH4 efflux (or, net ebullition), while a further 81 (6.9 %) showed very high rates of net CH4 uptake. The latter we termed "ebullition-driven CH4 uptake," due to the fact

that very high rates of CH4 uptake were observed following a putative bubble event. For these data, ebullition fluxes were calculated in one of two ways: for chambers showing steep non-linear increases, we fitted the data to a quadratic regression equation (P < 0.05), and fluxes were determined from the steep initial rise in CH4 concentrations. For chambers showing abrupt stochastic increases, fluxes were determined by calculating the total CH4 production over the course of the bubble event, in-line with the approach used by the investigators in other studies (Teh et al., 2011). We decided to omit these data from the final dataset because we could not exclude the possibility that these fluxes were caused by sampling effects, despite the care we took (e.g. physical distur-bance due to chamber placement or investigator movement; pressure effects caused by syringe sampling) led to ebullition. A summary of these ebullition data are presented in Supplementary Table 1 (below). We will revise the text to include a description of our data filtering procedure, and will also include the data shown in Supplementary Table 1, to provide more information to the reader on ebullition.

With respect to chamber placement; this was achieved by using a 2-m long pole to lower the flux chambers onto the water or saturated soil. Gas samples were collected with syringes using >2 m lengths of Tygon$^®$ tubing, after thoroughly purging the dead volumes in the sample lines. The text will be revised to provide these additional details on chamber placement and sampling technique.

5. Ebullition is relatively easy to detect e.g. if you start with very high concentrations or if you detect abrupt increases in CH4 concentration. Can you give us information on how you dealt which such data and how many of your chamber measurements were potentially affected by ebullition? Is there a correlation of flux strength and the poten-tial occurrence of ebullition? You also mention in your discussion that you measured occasionally high CH4 uptake values (l. 477). Is it possible that this was related to high CH4 concentrations at the beginning of chamber closure, potentially caused by ebullition? Did you find a correlation between initial CH4 concentration and calculated CH4 uptake values? I was also wondering whether the contrasting seasonal emission

patterns that you discuss (l. 417 and further) could be related to ebullition. As you see, I think you don't give use sufficient information about the potential occurrence of ebullition and you should clearly improve this.

Authors' response: Please see point 4 above. We did in fact see evidence of greater ebullition in higher emission environments. For example, ebullition was more common in Mixed Palm Swamp and M. flexuosa palm swamp (12.2 and 16.7 % of observations for those vegetation types, respectively), which are the two vegetation types that showed the highest CH4 fluxes. In contrast, forested (short pole) and forested vegetation, which showed the lowest CH4 fluxes, saw the lowest occurrence of ebullition (i.e. 6.9 and 10.5 % of observations, respectively). We also observed greater ebullition fluxes in the wet season, though the trend for ebullition-driven uptake was less clear. Due to the high variance in both ebullition and ebullition-driven uptake fluxes, we did not observe statistically significant differences in either of these rates among vegetation types, or between seasons. The text will be revised to include these data.

Regarding chambers that showed high oxidation rates; these high fluxes were in fact related to high initial concentrations, and we cannot exclude the possibility that these chambers could have been affected by ebullition, even if we did not see empirical evidence for this over the course of our chamber measurements (e.g. ebullition could have occurred immediately after chamber placement and before the first sample was taken). We will revise the discussion to recognise this possibility.

6. I have no problem with the fact that you measured negative N2O fluxes, since we see more and more evidence that globally this is quite an important process. However, some of the negative fluxes seem to be quite high in your figure 2. How do your N2O uptake rates compare to other published values and is it possible that this may also somehow related to ebullition? Also, here I would be interested if your negative values correlate with the initially measured concentration in your chambers. If ebullition played a role for negative Ch4 and N2O fluxes, you may expect that both strong CH4 uptake and N2O uptake would occur in the same chamber measurement. Did you check this?

Authors' response: Relative to other environments we have studied elsewhere in Peru (e.g. Kosñipata Valley, Manu National Park) and in the literature from upland environments, these uptake fluxes are very low; at least one order of magnitude lower than uptake fluxes observed in upland ecosystems (Teh et al., 2014). We believe it is highly unlikely that ebullition caused these trends, because we saw very little evidence of ebullition-driven N2O fluxes (only 3 out of 1181 observations, or 0.3 %), and these data were filtered to remove these three observations from the analysis presented here. We also saw no evidence that strong CH4 uptake was correlated with N2O uptake.

7. You probably measured your fluxes at different time of the day. Did you find any diurnal pattern in GHG fluxes? If yes, what could explain these observed differences and how would this affect your estimate of emission strength?

Authors' response: We did in fact conduct a study to investigate if gas fluxes showed evidence of diurnal variability, but did not find strong evidence of diurnal trends in fluxes. We will discuss this in the revised version of the manuscript.

8. The potential role of CH4 oxidation is remarkable absent from your discussion. Why is that the case? Do you think this is not important for the total measured CH4 fluxes?

Authors' response: For sake of brevity, we did not go discuss the potential role of gross CH4 oxidation in modulating net CH4 efflux. This is not because we did not believe gross CH4 oxidation was unimportant; rather, it was because we did not have the tools or the experimental design to make clear inferences about what proportion of produced CH4 was consumed prior to atmospheric egress. We do in fact believe that gross CH4 oxidation is very important, as demonstrated by past work in tropical systems that have used isotope tracers to deconvolve gross CH4 production and oxidation fluxes (von Fischer and Hedin, 2002;von Fischer and Hedin, 2007;Teh et al., 2005). For example, work by the lead author has demonstrated that gross methanotrophy may consume upwards of 48 % of produced CH4 in tropical soils (Teh et al., 2005). Follow-up experiments at these study sites could explore this question in the future. We will

revise the manuscript to include this information.

9 The version of the map in Fig. 1 that I saw did not have a very good quality. Can you provide a map where the four vegetation types that you used for stratification are included? The impression of Fig. 1 is that the total peat area is larger than what can be seen in the map. Can you adjust Fig. 1 in such a way that we see all of (or most of) the PMFB peatland area?

Authors' response: In the revised manuscript, we will provide a higher quality image than the one shown here, and will endeavour to include the information that the referee has requested.

REFERENCES Lahteenoja, O., Ruokolainen, K., Schulman, L., and Alvarez, J.: Amazonian floodplains harbour minerotrophic and ombrotrophic peatlands, Catena, 79, 140-145, 10.1016/j.catena.2009.06.006, 2009. Pangala, S. R., Moore, S., Hornibrook, E. R. C., and Gauci, V.: Trees are major conduits for methane egress from tropical forested wetlands, New Phytologist, 197, 524-531, 10.1111/nph.12031, 2013. Strack, M., Kellner, E., and Waddington, J. M.: Dynamics of biogenic gas bubbles in peat and their effects on peatland biogeochemistry, Global Biogeochemical Cycles, 19, n/a-n/a, 10.1029/2004GB002330, 2005. Teh, Y. A., Silver, W. L., and Conrad, M. E.: Oxygen effects on methane production and oxidation in humid tropical forest soils, Global Change Biology, 11, 1283-1297, 10.1111/j.1365-2486.2005.00983.x, 2005. Teh, Y. A., Silver, W. L., Sonnentag, O., Detto, M., Kelly, M., and Baldocchi, D. D.: Large Greenhouse Gas Emissions from a Temperate Peatland Pasture, Ecosystems, 14, 311-325, 10.1007/s10021-011-9411-4, 2011. Teh, Y. A., Diem, T., Jones, S., Huaraca Quispe, L. P., Baggs, E., Morley, N., Richards, M., Smith, P., and Meir, P.: Methane and nitrous oxide fluxes across an elevation gradient in the tropical Peruvian Andes, Biogeosciences, 11, 2325-2339, 10.5194/bg-11-2325-2014, 2014. von Fischer, J., and Hedin, L.: Separating methane production and consumption with a field-based isotope dilution technique., Global Biogeochemical Cycles, 16, 1-13, 10.1029/2001GB001448, 2002. von Fischer, J. C., and Hedin, L. O.: Controls on soil methane fluxes: Tests of

biophysical mechanisms using stable isotope tracers, Global Biogeochemical Cycles, 21, 9, Gb2007 10.1029/2006gb002687, 2007.

Please also note the supplement to this comment:
http://www.biogeosciences-discuss.net/bg-2017-48/bg-2017-48-AC1-supplement.pdf

[Figure]

**Supplementary Table 1.** Table displaying the proportion of observation in each vegetation type that showed evidence of ebullition, mean rates of ebullition and ebullition-driven $CH_4$ uptake. Values represent means and standard errors.

| Vegetation Type | Percentage of observations | Ebullition ($mg\ CH_4\text{-}C\ m^{-2}\ d^{-1}$) | | Ebullition-driven uptake ($mg\ CH_4\text{-}C\ m^{-2}\ d^{-1}$) | |
|---|---|---|---|---|---|
| | (%) | Wet Season | Dry Season | Wet Season | Dry Season |
| Forested | 10.5 | N/A | N/A | N/A | -136.4 ± 0.1 |
| Forested (short pole) | 6.9 | 994.6 ± 293.2 | 512.5 ± 153.0 | -95.8 ± 0.0 | -245.5 ± 48.9 |
| *M. flexuosa* Palm Swamp | 16.7 | 1192.0 ± 305.7 | 994.3 ± 237.3 | -869.4 ± 264.8 | -401.4 ± 59.9 |
| Mixed Palm Swamp | 12.2 | N/A | 733.6 ± 313.1 | N/A | -464.4 ± 565.9 |

**Fig. 1.**

---

## Author Comment (AC2) · 20 Mar 2017

10. The investigations by Teh et al. on CH4 and N2O emissions from tropical peatlands are recently of major interest. Particularly for the study region in the Amazon basin, knowledge on magnitude, pattern and controls of greenhouse gas fluxes is scarce. As this region is a potentially huge source of CH4, it is important to close this knowledge gap. This study could contribute to this process.

Authors' response: The authors would like to thank the referee for his/her kind and very supportive remarks.

11. The findings of large CH4 and negligible N2O emissions might have been antic-

ipated while the asynchronous CH4 fluxes with higher fluxes during the dry season for two of the four investigated vegetation types might not. Potential explanations for this result are discussed. However, there is plenty of literature on mechanisms and controls of peatland CH4 and N2O fluxes but appropriate references are missing in the introduction as well as in the discussion part. For example, it has been reported that CH4 fluxes do not increase or even decline when sites are inundating and that highest emissions occur for water levels close to the surface (e.g. Couwenberg et al., 2011, Hydrobiologia 674, 67-89).

Authors' response: As discussed in our response to the first referee (see point 2), we will revise the text to include a more through discussion of the underlying controls on CH4 and N2O flux. We also thank the referee for the suggested reference, and will incorporate the findings from this work into the new version of the manuscript.

12. Furthermore, I strongly recommend to revise the abstract and the presentation of results. The abstract mainly lists the results but doesn't tell anything about motivation, objective and main conclusions of the study. The two figures are of poor quality and Fig. 2 is not very helpful for interpretation of results due to the huge amounts of outliers. I would recommend to rather show columns with standard errors or maxima/minima. And why was the data shown in the figure grouped for vegetation type but not for different seasons? Also, figures that visualize the relationships between GHG fluxes and the measured environmental variables would be interesting. Although the relationships might be very weak, this would give the reader a better idea of the dataset.

Authors' response: Thank you for these suggestions. We will revise the abstract and figure 2 along the lines suggested here. With respect to seasonal trends, we made the decision to show this information in Table 2 rather than as a figure showing seasonal differences. Finally, with respect to the relationship between GHG fluxes and environmental variables (e.g. scatterplots of environmental variables versus gas flux), we can incorporate some of this information into appendices or as online supplementary materials for the revised version of the text.

13. P4, L60: Peatlands are not necessarily peat-forming. In contrast, many peatlands have been drained for utilization purposes which turns them into significant C sources and in regions like Central Europe, only a few percent of peatlands are still in a natural condition and thus peat-forming. Please be more specific about the state of peatlands in the study region.

Authors' response: The peatlands in the PMFB are unmanaged and have not been affected by human disturbance, unlike their counterparts in SE Asia.

14. P4, L61-65: Are you talking about peatlands in general or about peatlands in the Amazon basin? If you mean peatlands in general, your statements are not correct as there are several studies on peatland GHG emissions but most of them were conducted in the boreal or temperate zone.

Authors' response: We were referring to peatlands in the Amazon basin; the focus of past research in the region has been on mineral soil wetlands.

15. P5, L90: I wouldn't classify a peat depth of 3.9 m as shallow. Following the international definition, peatlands are defined by a minimum of 20 cm of peat deposit, which could be classified as shallow. It seems that peat deposits in the study region are all very deep.

Authors' response: In the revised version of this text, we will correct the language so as to reflect this point.

16. P7, L127-128: You do not mention the months of September and October. Are they neither wet season nor dry season?

Authors' response: September and October represent a transitional period between the wet and dry seasons.

17. P7, L135: "Pure peat" is not a soil classification! Please use World reference base (2015) to classify your sites. Qualifiers should be used to characterize the soils more precisely.

Authors' response: The referee's concern is duly noted and the revised version of the text will be changed according to the referee's suggestion.

18. P7, L136: The pH values given are not in line with Table 2. Are these values from preliminary measurements?

Authors' response: The values reported on line 136 represent the range of values observed in prior studies, whereas the values reported in Table 2 reflect more specific data from our study plots. We will make this clear in the revised manuscript.

19. P8, L150: The number of plots remains unclear to me. You mention 229 plots but the numbers below summarize to 148 plots and in Table 1 you list 161 plots. Please clarify. Authors' response: The total number of plots should be 161, in accordance with Table 1. The text will be corrected accordingly.

20. P9, L177: Please clarify the procedure of the measurement campaigns. How long were these campaigns, did you sample each plot only once, several times per day or also on different days?

Authors' response: The duration of the campaign for each study site varied depending on its size. Each study site was generally sampled only once for each campaign.

21. P9, L178: What about the wet season in 2013? Why didn't you measure the fluxes during that period?

Authors' response: Due to personal circumstances, we were unable to collect data during the 2013 wet season.

22. P9, L186: How is it possible to place the chamber in a distance of 2 m? I cannot imagine how this practically works. And what about the sampling procedure? You have to get quite close to the chambers for that. Please clarify.

Authors' response: As discussed in our response to the first referee (see point 4), this was achieved by using a 2-m long pole to lower the flux chambers onto the water or

saturated soil. Gas samples were collected with syringes using >2 m lengths of Tygon[®] tubing, after thoroughly purging the dead volumes in the sample lines. If this paper is accepted for publication, we will revise the methods to provide these additional details on chamber placement and sampling technique.

23. P10, L199: Does it mean that fluxes were calculated from linear or non-linear regressions depending on the individual concentration trend against time? It is important to clarify this as linear regression can lead to substantial underestimation of fluxes as a consequence of decreasing concentration gradients over time. And which quality criteria have been used to ensure the reliability of computed fluxes?

Authors' response: The referee is correct; the manuscript will be revised to clarify this point.

24. P13, L268: The paragraphs for the results of different variables always have the same wording, which gives a quite uninspired impression.

Authors' response: We strove for clarity of expression in this section of the text, and believe that this reporting format achieves this goal.

25. P14, L281: I don't find it very meaningful to do statistics on measurements of air temperature. Also, you would have to compare exactly the same periods, otherwise the results could be misleading.

Authors' response: Air temperature measurements can be useful because they provide an indication of overall climatic variability, not only between seasons but among ecosystems, which may have different amounts of canopy closure. We have therefore provided this information to provide the reader a sense of overall patterns in climate variability among study sites. However, if the referee strongly objects to incluson of these data, we can remove it from the revised version of the manuscript.

26. P16, L325: Several different statistical tests were applied but not mentioned in the statistical analyses section.

Authors' response: We did not specifically mention the Wilcoxon signed-rank test in the statistics section; however, we did indicate that non-parametric tests were used under certain circumstances (see line 222). The text will be revised to provide specific detail on what non-parametric tests were employed.

27. P19, L398: It has to be considered that conclusions on that can only be drawn when annual CH4 budgets can be estimated from regular or automatic measurements in high temporal resolution.

Authors' response: The sentence referred to by the referee includes a qualifier (i.e. "may be") to denote that we believe that it is highly likely that this region is an important regional contributor to CH4 flux, but do not necessarily claim that this is entirely proven. Although we agree with the referee that regular or higher frequency measurements would be required to reach a more definitive conclusion, we believe that the weight of evidence supports our qualitative claim, including findings not only from this paper but from regional atmospheric sampling studies (Wilson et al., 2016).

28. P20, L 416: The water tables of the studied sites do in my opinion not allow the definition of oxic conditions as the water tables reported were quite high even in the dry season. Particularly non-degraded peat has a high water holding capacity, thus also when the water table drops below the soil surface, water filled pore space remains high in the top soil, still preventing CH4 oxidation.

Authors' response: The sentence referred to by the referee includes a qualifier (i.e. "more") to indicate that we are comparing oxygen availability during the wet and dry season, rather than making a statement about absolute oxygen content. The data provided in Table 2 demonstrate that most of the vegetation types, with the exception of Mixed Palm Swamp, show elevated dissolved oxygen levels during the dry season, supporting the idea that the soils contained more oxygen than during the wet season.

29. P21, L 438: No references are given in this section. The weak relationship is probably a result of the overall high water levels as the general assumption that CH4

emissions increase with water level is based on measurements from sites with huge drainage gradients. Generally, CH4 emissions increase exponentially when the water level passes a threshold of roughly 20 – 30 cm below ground. For water levels close to or above the surface, CH4 fluxes are often extremely variable. Some references would be very helpful here for the interpretation of your results. Also, methodological issues should be discussed as CH4 emissions mainly occur in terms of erratic ebullition when water tables are above the soil surface. This might be difficult to detect with small chambers during a short period of enclosure.

Authors' response: We will include additional references to this section in order to ensure that our statements are more clearly supported by prior research (Couwenberg et al., 2010;Couwenberg et al., 2011). Moreover, we will revise the text to include greater discussion about ebullition, in-line with the first referee's remarks.

30. P23, L479: Where negative CH4 fluxes also measured for water tables above ground? This would be rather unreliable in my opinion as one would not expect CH4 uptake in water saturated soil or even open water. Also, I assume that there should be more recent literature on CH4 exchange in tropical peatlands.

Authors' response: No negative CH4 fluxes were observed when water tables were above the soil surface, only when water tables were below the soil surface.

31. P24, L505-508: It is for me very unlikely that the different ecosystems do not differ in N availability. This conclusion cannot be drawn from equally low N2O emissions as there are probably other reasons for negligible N2O fluxes also in the nutrient-rich ecosystems. For example, there might be a higher N uptake by productive plant species at the nutrient-rich sites, competing with the potentially N2O producing microbes or N compounds are completely reduced to N2 during denitrification because of permanently anoxic conditions. Technical corrections P2, L9: The numbering of sections starts with 2.

Authors' response: We do not disagree with the referee; we simply indicated that this

may be one possible explanation for the trends in our data.

32. P5, L99: Replace "positive water tables" by "high water tables" or "water tables above ground".

Authors' response: This editorial suggestion will be taken in the revised version of the text.

33. P10, L197: I assume that the instrumental precision was > 95 % or the instrumental noise was < 5 %.

Authors' response: Yes.

34. P10, L210: In which height was the air temperature measured?

Authors' response: Approximately 1.3 m above the surface.

35. P10, L211: Please add manufacturer.

Authors' response: This editorial suggestion will be taken in the revised version of the text.

36. P13, L270: add "different" after "significantly".

Authors' response: This editorial suggestion will be taken in the revised version of the text.

37. P14, L282: "Soil temperature" has to be replaced by "air temperature"

Authors' response: This editorial suggestion will be taken in the revised version of the text.

38. P15, L314: Add "during" before "the dry season".

Authors' response: This editorial suggestion will be taken in the revised version of the text.

39. P18, L364: Plural: relationships

Authors' response: This editorial suggestion will be taken in the revised version of the text.

40. P18, L372: Results should not be interpreted in the "Results" section.

Authors' response: The text will be revised to take into account the referee's concern.

41. P18, L382: Add "electrical" before "conductivity".

Authors' response: This editorial suggestion will be taken in the revised version of the text.

42. P18, L383: Why do you mention CO2 here? No results on CO2 were shown.

Authors' response: Reference to CO2 will be removed in the revised version of the text.

43. P19, L394: Please round up to 1510.

Authors' response: This editorial suggestion will be taken in the revised version of the text.

44. P25, L526: Replace "these data" by "our data".

Authors' response: This editorial suggestion will be taken in the revised version of the text.

REFERENCES Couwenberg, J., Dommain, R., and Joosten, H.: Greenhouse gas fluxes from tropical peatlands in south-east Asia, Global Change Biology, 16, 1715-1732, 10.1111/j.1365-2486.2009.02016.x, 2010.   Couwenberg, J., Thiele, A., Tanneberger, F., Augustin, J., Bärisch, S., Dubovik, D., Liashchynskaya, N., Michaelis, D., Minke, M., Skuratovich, A., and Joosten, H.: Assessing greenhouse gas emissions from peatlands using vegetation as a proxy, Hydrobiologia, 674, 67-89, 10.1007/s10750-011-0729-x, 2011. Wilson, C., Gloor, M., Gatti, L. V., Miller, J. B., Monks, S. A., McNorton, J., Bloom, A. A., Basso, L. S., and Chipperfield, M. P.: Contribution of regional sources to atmospheric methane over the Amazon Basin in

2010 and 2011, Global Biogeochem. Cycles, 30, 400–420, 10.1002/2015GB005300, 2016.

Please also note the supplement to this comment:
http://www.biogeosciences-discuss.net/bg-2017-48/bg-2017-48-AC2-supplement.pdf

---

## Author Response (AR1)

**DETAILED RESPONSE TO REFEREES**

On behalf of my co-authors, I would like to thank the Associate Editor and the two anonymous referees for their thoughtful and constructive comments on our manuscript. Please find enclosed a revised version of the text, where we have sought to address all the referees' concerns, in-line with the responses that we have provided to the referees during the discussion period for the paper. A detailed description of how we have responded to the referees comments is provided below.

**RESPONSE TO REFEREE 1**

**1.** *Although the number of studies on trace gas fluxes from lowland tropical peatlands are steadily increasing, measurements are completely absent from some important large peatland areas. The peats in the foreland basin of Pastaza Maranon (PMFB) in Peru is an example of an important tropical peatland area from which we have no empirical studies on CH4 and N2O fluxes. As such, this study is timely and novel. The authors conducted chamber based CH4 and N2O fluxes stratified according to the four dominant vegetation types ranging from nutrient rich floodplains to nutrient poor bogs. While the spatial resolution of the sampling is good (see Table 1) the temporal resolution is restricted to four time sampling over a period of two years (sampling twice during the wet season and twice during a dry season). However, given the difficult accessibility and the fact that this is the first report on fluxes from this important peat area this is acceptable for this initial study. Furthermore, we do learn important controls since supporting variables were measured as well. Nevertheless if have several point that should be addressed before the manuscript can be accepted for publication.*

**Authors' response:** The authors would like to thank the referee for his/her thoughtful and insightful comments on our manuscript. We welcome this opportunity to improve the manuscript for the wider readership of *Biogeosciences*, and hope that the changes we have produced will meet with your satisfaction..

**2.** *In your introduction, I miss a section on the known controls of CH4 and N2O fluxes in peatlands. I think it is important to include this, since it is otherwise unclear why you*

*measured the environmental variables that you did. Are there some potentially important factors that you did not measure and if so, why was that the case?*

**Authors' response:** In the revised version of the manuscript, the introduction has been modified to include a more thorough description of the controls on $CH_4$ and $N_2O$ flux (lines 150-196).

With respect to potentially important factors that we did not measure: we did not quantify $CH_4$ emissions from woody plants nor did we specifically seek to quantify ebullition. While plant-mediated fluxes are believed to be important in tropical wetland ecosystems (Pangala et al., 2013), we did not have enough data on floristic composition or individual plant identities within our plots to come-up with a sampling design that would adequately represent plant-mediated fluxes from our study sites. Likewise, ebullition is often characterized by high spatial and temporal variability. In order to develop representative measures of ebullition, we would have to use quasi-continuous, automated flux chambers or an inverted "flux funnel" approach (Strack et al., 2005). However, we lacked the resources to implement either of these approaches in this study. In the revised text, we have expanded the discussion of ebullition in order to meet the concerns raised in points 4 and 5 (see below). We have also revised the text to include an expanded discussion of our study's limitations in the discussion (lines 780-830).

For $N_2O$, even though inorganic N is thought to be one of the major drivers of $N_2O$ flux, we did not collect data on inorganic N ($NH_4^+$, $NO_3^-$), because the relatively modest budget for this project did not accommodate costs for inorganic N analysis.

**3.** *In l.142 and l.147 you mention soil Ca, it remains a mystery in which form this Ca occurs. Please provide more details, is Ca an important environmental control on CH4 or N2O fluxes?*

**Authors' response:** The Ca in these systems occurs as dissolved inorganic $Ca^{2+}$ associated with the soil or peat exchange complex, or Ca found in secondary minerals (Lahteenoja et al., 2009;Lahteenoja and Page, 2011). As far as we are aware, Ca has no direct effect on $CH_4$

or N$_2$O fluxes, although Ca may indirectly influence trace gas exchange by influencing plant productivity and organic matter decay. We have revised the text to clarify this point (lines 264-272).

**4.** *You describe your chambers as floating static chambers (l. 179). You furthermore write that chamber were placed on the soil surface from a distance of no closer than 2m to reduce the risk of ebullition (l. 186). For me it is hard to believe that ebullition was completely excluded and I also cannot understand how you can place these chambers at a distance of 2 m? Nobody has arms that long (at least I don't), so how was this done in detail? And, especially, how did you take the gas samples from your chambers using syringes without causing ebullition?*

**Authors' response:** The referee is correct that we were unable to entirely exclude ebullition from our dataset. We did in fact find evidence of ebullition, with 164 of the 1181 chamber observations (13.9 %) showing signs of ebullition (e.g. abrupt, non-linear changes in headspace concentrations). Of these 164 observations, 83 (7.0 %) showed net CH$_4$ efflux (or, net ebullition), while a further 81 (6.9 %) showed very high rates of net CH$_4$ uptake. The latter we termed "ebullition-driven CH$_4$ uptake," due to the fact that very high rates of CH$_4$ uptake were observed following a putative bubble event. For these data, ebullition fluxes were calculated in one of two ways: for chambers showing steep non-linear increases, we fitted the data to a quadratic regression equation ($P < 0.05$), and fluxes were determined from the steep initial rise in CH$_4$ concentrations. For chambers showing abrupt stochastic increases, fluxes were determined by calculating the total CH$_4$ production over the course of the bubble event, in-line with the approach used by the investigators in other studies (Teh et al., 2011).  The text has now been amended to include a more thorough description of how these ebullition data were handled and interpreted in the methods (lines 377-400), results and discussion (lines 432-453, lines 456-458, lines 524-567, lines 661-706, Table 2).

With respect to chamber placement; this was achieved by using a 2-m long pole to lower the flux chambers onto the water or saturated soil. Gas samples were collected with syringes using >2 m lengths of Tygon® tubing, after thoroughly purging the dead volumes in the sample lines. The text has now been revised to provide these additional details on chamber placement and sampling technique (lines 358-362).

**5.** *Ebullition is relatively easy to detect e.g. if you start with very high concentrations or if you detect abrupt increases in CH4 concentration. Can you give us information on how you dealt which such data and how many of your chamber measurements were potentially affected by ebullition? Is there a correlation of flux strength and the potential occurrence of ebullition? You also mention in your discussion that you measured occasionally high CH4 uptake values (l. 477). Is it possible that this was related to high CH4 concentrations at the beginning of chamber closure, potentially caused by ebullition? Did you find a correlation between initial CH4 concentration and calculated CH4 uptake values? I was also wondering whether the contrasting seasonal emission patterns that you discuss (l. 417 and further) could be related to ebullition. As you see, I think you don't give use sufficient information about the potential occurrence of ebullition and you should clearly improve this.*

**Authors' response:** Please see point 4 above. We did in fact see evidence of greater ebullition in higher emission environments. For example, ebullition was more common in Mixed Palm Swamp and *M. flexuosa* palm swamp (12.2 and 16.7 % of observations for those vegetation types, respectively), which are the two vegetation types that showed the highest $CH_4$ fluxes. In contrast, forested (short pole) and forested vegetation, which showed the lowest $CH_4$ fluxes, saw the lowest occurrence of ebullition (i.e. 6.9 and 10.5 % of observations, respectively). We also observed greater ebullition fluxes in the wet season, though the trend for ebullition-driven uptake was less clear. Due to the high variance in both ebullition and ebullition-driven uptake fluxes, we did not observe statistically significant differences in either of these rates among vegetation types, or between seasons. The manuscript has now been revised to incorporate this information (lines 377-400, lines 432-453, lines 456-458, lines 524-567, lines 661-706, Table 2).

Regarding chambers that showed high oxidation rates; these high fluxes were in fact related to high initial concentrations, and we cannot exclude the possibility that these chambers could have been affected by ebullition, even if we did not see empirical evidence for this over the course of our chamber measurements (e.g. ebullition could have occurred immediately after chamber placement and before the first sample was taken). The text has now been revised to incorporate this information (lines 798-830).

**6.** *I have no problem with the fact that you measured negative N2O fluxes, since we see more and more evidence that globally this is quite an important process. However, some of the negative fluxes seem to be quite high in your figure 2. How do your N2O uptake rates compare to other published values and is it possible that this may also somehow related to ebullition? Also, here I would be interested if your negative values correlate with the initially measured concentration in your chambers. If ebullition played a role for negative Ch4 and N2O fluxes, you may expect that both strong CH4 uptake and N2O uptake would occur in the same chamber measurement. Did you check this?*

**Authors' response:** Relative to other environments we have studied elsewhere in Peru (e.g. Kosñipata Valley, Manu National Park) and in the literature from upland environments, these uptake fluxes are very low; at least one order of magnitude lower than uptake fluxes observed in upland ecosystems (Teh et al., 2014). We believe it is highly unlikely that ebullition caused these trends, because we saw very little evidence of ebullition-driven $N_2O$ fluxes (only 3 out of 1181 observations, or 0.3 %), and these data were filtered to remove these three observations from the analysis presented here. We also saw no evidence that strong $CH_4$ uptake was correlated with $N_2O$ uptake. The text has been amended to acknowledge that we saw only limited evidence of $N_2O$ ebullition (lines 456-458).

**7.** *You probably measured your fluxes at different time of the day. Did you find any diurnal pattern in GHG fluxes? If yes, what could explain these observed differences and how would this affect your estimate of emission strength?*

**Authors' response:** We did in fact conduct a subsidiary study to investigate if gas fluxes showed evidence of diurnal variability, but did not find strong evidence of diurnal trends in fluxes. The text has now been revised to provide this information (lines 348-351 and lines 529-530).

**8.** *The potential role of CH4 oxidation is remarkable absent from your discussion. Why is that the case? Do you think this is not important for the total measured CH4 fluxes?*

**Authors' response:** For sake of brevity, we did not go discuss the potential role of gross $CH_4$ oxidation in modulating net $CH_4$ efflux. This is not because we did not believe gross $CH_4$ oxidation was unimportant; rather, it was because we did not have the tools or the experimental design to make clear inferences about what proportion of produced $CH_4$ was consumed prior to atmospheric egress. We do in fact believe that gross $CH_4$ oxidation is very important, as demonstrated by past work in tropical systems that have used isotope tracers to deconvolve gross $CH_4$ production and oxidation fluxes (von Fischer and Hedin, 2002;von Fischer and Hedin, 2007;Teh et al., 2005). For example, work by the lead author has demonstrated that gross methanotrophy may consume upwards of 48 % of produced $CH_4$ in tropical soils (Teh et al., 2005). Follow-up experiments at these study sites could explore this question in the future. We will revise the manuscript to include this information.

**9** *The version of the map in Fig. 1 that I saw did not have a very good quality. Can you provide a map where the four vegetation types that you used for stratification are included? The impression of Fig. 1 is that the total peat area is larger than what can be seen in the map. Can you adjust Fig. 1 in such a way that we see all of (or most of) the PMFB peatland area?*

**Authors' response:** Figure 1 has now been revised to better illustrate the distribution of peat-rich areas across the study site.

**RESPONSE TO REFEREE 2**

**10.** *The investigations by Teh et al. on CH4 and N2O emissions from tropical peatlands are recently of major interest. Particularly for the study region in the Amazon basin, knowledge on magnitude, pattern and controls of greenhouse gas fluxes is scarce. As this region is a potentially huge source of CH4, it is important to close this knowledge gap. This study could contribute to this process.*

**Authors' response:** The authors would like to thank the referee for his/her kind and very supportive remarks.

**11.** *The findings of large CH4 and negligible N2O emissions might have been anticipated while the asynchronous CH4 fluxes with higher fluxes during the dry season for two of the four investigated vegetation types might not. Potential explanations for this result are discussed. However, there is plenty of literature on mechanisms and controls of peatland CH4 and N2O fluxes but appropriate references are missing in the introduction as well as in the discussion part. For example, it has been reported that CH4 fluxes do not increase or even decline when sites are inundating and that highest emissions occur for water levels close to the surface (e.g. Couwenberg et al., 2011, Hydrobiologia 674, 67-89).*

**Authors' response:** Please see point 2 above. We also thank the referee for the suggested reference, and have incorporated the findings from this work into the new version of the manuscript.

**12.** *Furthermore, I strongly recommend to revise the abstract and the presentation of results. The abstract mainly lists the results but doesn't tell anything about motivation, objective and main conclusions of the study. The two figures are of poor quality and Fig. 2 is not very helpful for interpretation of results due to the huge amounts of outliers. I would recommend to rather show columns with standard errors or maxima/minima. And why was the data shown in the figure grouped for vegetation type but not for different seasons? Also, figures that visualize the relationships between GHG fluxes and the measured environmental*

*variables would be interesting. Although the relationships might be very weak, this would give the reader a better idea of the dataset.*

**Authors' response:** Thank you for these suggestions. The abstract (lines 10-60) and Figure 2 have now been revised in-line with the referee's suggestions. With respect to seasonal trends, we made the decision to show this information in a table rather than as a figure to show seasonal differences (see Table 4 in the revised text). We believe that a tabular format shows seasonal trends with greater clarity than a more complex figure showing both site and seasonal differences.

With respect to the relationship between GHG fluxes and environmental variables (e.g. scatterplots of environmental variables versus gas flux), we have now generated scatter plots for some of the key variables discussed in the results section (see Supplementary Online Materials Figures S1 to S4).

**13.** *P4, L60: Peatlands are not necessarily peat-forming. In contrast, many peatlands have been drained for utilization purposes which turns them into significant C sources and in regions like Central Europe, only a few percent of peatlands are still in a natural condition and thus peat-forming. Please be more specific about the state of peatlands in the study region.*

**Authors' response:** The peatlands in the PMFB are unmanaged and have not been affected by human disturbance, unlike their counterparts in SE Asia.In the wider Amazon basin, only peatlands in the Madre de Dios region have been heavily affected by human activity. The revised text now includes this additional information (lines 98-102).

**14.** *P4, L61-65: Are you talking about peatlands in general or about peatlands in the Amazon basin? If you mean peatlands in general, your statements are not correct as there are several studies on peatland GHG emissions but most of them were conducted in the boreal or temperate zone.*

**Authors' response:** We were referring to peatlands in the Amazon basin; the focus of past research in the region has been on mineral soil wetlands. The text has now been revised to clarify this point (lines 102-105).

**15.** *P5, L90: I wouldn't classify a peat depth of 3.9 m as shallow. Following the international definition, peatlands are defined by a minimum of 20 cm of peat deposit, which could be classified as shallow. It seems that peat deposits in the study region are all very deep.*

**Authors' response:** The text has now been revised so that we simply refer to the depth range of the peat (lines 132-133).

**16.** *P7, L127-128: You do not mention the months of September and October. Are they neither wet season nor dry season?*

**Authors' response:** September and October represent a transitional period between the wet and dry seasons, and the text has now been revised to clarify this point (lines 226-228).

**17.** *P7, L135: "Pure peat" is not a soil classification! Please use World reference base (2015) to classify your sites. Qualifiers should be used to characterize the soils more precisely.*

**Authors' response:** The referee's concern is duly noted, and we have now revised the text accordingly (lines 235-272).

**18.** *P7, L136: The pH values given are not in line with Table 2. Are these values from preliminary measurements?*

**Authors' response:** The values reported on line 136 represent the range of values observed in prior studies, whereas the values reported in Table 2 reflect more specific data from our study plots. We have made this clearer in the revised version of the text (lines 237-239).

**19.** *P8, L150: The number of plots remains unclear to me. You mention 229 plots but the numbers below summarize to 148 plots and in Table 1 you list 161 plots. Please clarify.*

**Authors' response:** The total number of plots should be 239, with the following breakdown by vegetation type:

Forested: 21 plots
Forested (short pole): 47 plots
M. Flexuosa palm swamp: 153 plots
Mixed palm swamp: 18 plots

The revised version of the text (lines 313-320) and Table 1 have been corrected accordingly.

**20.** *P9, L177: Please clarify the procedure of the measurement campaigns. How long were these campaigns, did you sample each plot only once, several times per day or also on different days?*

**Authors' response:** The duration of the campaign for each study site varied depending on its size. Each study site was generally sampled only once for each campaign. The revised text has been updated accordingly (lines 348-351).

**21.** *P9, L178: What about the wet season in 2013? Why didn't you measure the fluxes during that period?*

**Authors' response:** Due to personal circumstances, we were unable to collect data during the 2013 wet season.

**22.** *P9, L186: How is it possible to place the chamber in a distance of 2 m? I cannot imagine how this practically works. And what about the sampling procedure? You have to get quite close to the chambers for that. Please clarify.*

**Authors' response:** Please see point 4 above.

**23.** *P10, L199: Does it mean that fluxes were calculated from linear or non-linear regressions depending on the individual concentration trend against time? It is important to clarify this as linear regression can lead to substantial underestimation of fluxes as a consequence of decreasing concentration gradients over time. And which quality criteria have been used to ensure the reliability of computed fluxes?*

**Authors' response:** The referee is correct; the revised manuscript has been altered to clarify this point (lines 377-400).

**24.** *P13, L268: The paragraphs for the results of different variables always have the same wording, which gives a quite uninspired impression.*

**Authors' response:** We strove for clarity of expression in this section of the text, and believe that this reporting format achieves this goal.

**25.** *P14, L281: I don't find it very meaningful to do statistics on measurements of air temperature. Also, you would have to compare exactly the same periods, otherwise the results could be misleading.*

**Authors' response:** Air temperature measurements can be useful because they provide an indication of overall climatic variability, not only between seasons but among ecosystems, which may have different amounts of canopy closure. We have therefore provided this information to provide the reader a sense of overall patterns in climate variability among study sites.

**26.** *P16, L325: Several different statistical tests were applied but not mentioned in the statistical analyses section.*

**Authors' response:** We did not specifically mention the Wilcoxon signed-rank test in the statistics section; however, we did indicate that non-parametric tests were used under certain circumstances (lines 422-423). The revised text has also been update to provide specific detail on what non-parametric tests were employed (line 423).

**27.** *P19, L398: It has to be considered that conclusions on that can only be drawn when annual CH4 budgets can be estimated from regular or automatic measurements in high temporal resolution.*

**Authors' response:** The sentence referred to by the referee includes a qualifier (i.e. "may be") to denote that we believe that it is highly likely that this region is an important regional contributor to $CH_4$ flux, but do not necessarily claim that this is entirely proven. Although we agree with the referee that regular or higher frequency measurements would be required to reach a more definitive conclusion, we believe that the weight of evidence supports our qualitative claim, including findings not only from this paper but from regional atmospheric sampling studies (Wilson et al., 2016).

**28.** *P20, L 416: The water tables of the studied sites do in my opinion not allow the definition of oxic conditions as the water tables reported were quite high even in the dry season. Particularly non-degraded peat has a high water holding capacity, thus also when the water table drops below the soil surface, water filled pore space remains high in the top soil, still preventing CH4 oxidation.*

**Authors' response:** The sentence referred to by the referee includes a qualifier (i.e. "more") to indicate that we are comparing oxygen availability during the wet and dry season, rather than making a statement about absolute oxygen content. The data provided in Table 2 demonstrate that most of the vegetation types, with the exception of Mixed Palm Swamp, show elevated dissolved oxygen levels during the dry season, supporting the idea that the soils contained more oxygen than during the wet season.

**29.** *P21, L 438: No references are given in this section. The weak relationship is probably a result of the overall high water levels as the general assumption that CH4 emissions increase with water level is based on measurements from sites with huge drainage gradients. Generally, CH4 emissions increase exponentially when the water level passes a threshold of roughly 20 – 30 cm below ground. For water levels close to or above the surface, CH4 fluxes are often extremely variable. Some references would be very helpful here for the*

*interpretation of your results. Also, methodological issues should be discussed as CH4 emissions mainly occur in terms of erratic ebullition when water tables are above the soil surface. This might be difficult to detect with small chambers during a short period of enclosure.*

**Authors' response:** The text has been revised to include the point raised here and the additional citations suggested the referee (lines 724-738, Supplementary Online Materials Figure S2) (Couwenberg et al., 2010;Couwenberg et al., 2011). Moreover, the text has now been heavily revised to include a wider discussion of ebullition, in-line with the first referee's concerns (see points 4 and 5 above).

**30.** *P23, L479: Where negative CH4 fluxes also measured for water tables above ground? This would be rather unreliable in my opinion as one would not expect CH4 uptake in water saturated soil or even open water. Also, I assume that there should be more recent literature on CH4 exchange in tropical peatlands.*

**Authors' response:** No negative $CH_4$ fluxes were observed when water tables were above the soil surface, only when water tables were below the soil surface.

**31.** *P24, L505-508: It is for me very unlikely that the different ecosystems do not differ in N availability. This conclusion cannot be drawn from equally low N2O emissions as there are probably other reasons for negligible N2O fluxes also in the nutrient-rich ecosystems. For example, there might be a higher N uptake by productive plant species at the nutrient-rich sites, competing with the potentially N2O producing microbes or N compounds are completely reduced to N2 during denitrification because of permanently anoxic conditions. Technical corrections  P2, L9: The numbering of sections starts with 2.*

**Authors' response:** We do not disagree with the referee; we simply indicated that this may be one possible explanation for the trends in our data.

**32.** *P5, L99: Replace "positive water tables" by "high water tables" or "water tables above ground".*

**Authors' response:** Editorial suggestion taken.

**33.** *P10, L197: I assume that the instrumental precision was > 95 % or the instrumental noise was < 5 %.*

**Authors' response:** Yes.

**34.** *P10, L210: In which height was the air temperature measured?*

**Authors' response:** Approximately 1.3 m above the surface; the revised text has been modified accordingly (line 407).

**35.** *P10, L211: Please add manufacturer.*

**Authors' response:** Editorial suggestion taken.

**36.** *P13, L270: add "different" after "significantly".*

**Authors' response:** Editorial suggestion taken.

**37.** *P14, L282: "Soil temperature" has to be replaced by "air temperature"*

**Authors' response:** Editorial suggestion taken.

**38.** *P15, L314: Add "during" before "the dry season".*

**Authors' response:** Editorial suggestion taken.

**39.** *P18, L364: Plural: relationships*

**Authors' response:** Editorial suggestion taken.

**40.** *P18, L372: Results should not be interpreted in the "Results" section.*

**Authors' response:** Editorial suggestion taken; interpretive text was deleted.

**41.** *P18, L382: Add "electrical" before "conductivity".*

**Authors' response:** Editorial suggestion taken.

**42.** *P18, L383: Why do you mention CO2 here? No results on CO2 were shown.*

**Authors' response:** Editorial suggestion taken.

**43.** *P19, L394: Please round up to 1510.*

**Authors' response:** Editorial suggestion taken

**44.** *P25, L526: Replace "these data" by "our data".*

**Authors' response:** Editorial suggestion taken.

**REFERENCES**

[revised manuscript text omitted]

Values reported here are means and standard errors. Lower case letters indicate significant differences among vegetation types within the wet or dry season (Fisher's LSD, $P < 0.05$).

| Vegetation Type | Peat Temperature (°C) | | Air Temperature (°C) | | Conductivity ($\mu$S m$^{-2}$) | | Dissolved Oxygen (%) | | Water Table Level (cm) | | pH | |
|---|---|---|---|---|---|---|---|---|---|---|---|---|
| | Wet Season | Dry Season | Wet Season | Dry Season | Wet Season | Dry Season | Wet Season | Dry Season | Wet Season | Dry Season | Wet Season | Dry Season |
| Forested | 26.1 ± 0.1a | 24.7 ± 0.0a | 28.8 ± 0.7a | 26.4 ± 0.3a | 79.0 ± 5.9a | 75.9 ± 5.7a | 0.2 ± 0.1a | 18.9 ± 4.4a | 110.8 ± 9.3a | -13.2 ± 0.7a | 5.88 ± 0.15a | 6.31 ± 0.04a |
| Forested (short pole) | 25.2 ± 0.0b | 24.8 ± 0.1a | 27.6 ± 0.1b | 27.5 ± 0.1b | 21.0 ± 0.0b | 48.5 ± 4.8b | 4.4 ± 0.0a | 33.1 ± 2.6b | 26.9 ± 0.5b | -4.7 ± 0.4b | 4.88 ± 0.01b | 3.8 ± 0.03b |
| M. flexuosa Palm Swamp | 25.6 ± 0.6c | 25.3 ± 0.1b | 26.3 ± 0.1c | 26.4 ± 0.1a | 45.9 ± 2.1c | 51.9 ± 1.8b | 19.4 ± 1.3b | 17.3 ± 1.5a | 37.2 ± 1.7c | 6.1 ± 1.3c | 5.04 ± 0.03c | 5.49 ± 0.03c |
| Mixed Palm Swamp | 26.0 ± 0.0a | 25.0 ± 0.1ab | 26.1 ± 0.1c | 28.2 ± 0.3b | 100.0 ± 0.2d | 206.4 ± 4.2c | 0.0 ± 0.0a | 0.0 ± 0.0c | 183.7 ± 1.7d | -2.4 ± 0.3b | 6.1 ± 0.03a | 6.82 ± 0.02d |

**Table 4.** Trace gas fluxes for each vegetation type for the wet and dry season. Values reported here are means and standard errors. Upper case letters indicate significant differences in gas flux between seasons with a vegetation type, while lower case letters indicate significant differences among vegetation types within a season (Fisher's LSD, $P < 0.05$).

| Vegetation Type | Methane Flux ($mg\ CH_4\text{-}C\ m^{-2}\ d^{-1}$) | | Nitrous Oxide Flux ($\mu g\ N_2O\text{-}N\ m^{-2}\ d^{-1}$) | |
|---|---|---|---|---|
| | Wet Season | Dry Season | Wet Season | Dry Season |
| Forested | 6.7 ± 1.0Aa | 47.2 ± 5.4Ba | 2.54 ± 1.48 | -1.16 ± 1.20 |
| Forested (short pole) | 60.4 ± 9.1Ab | 18.8 ± 2.6Bb | 1.16 ± 0.54 | -0.42 ± 0.90 |
| *M. flexuosa* Palm Swamp | 46.7 ± 8.4Ac | 28.3 ± 2.6Bc | 1.14 ± 0.35 | 0.92 ± 0.61 |
| Mixed Palm Swamp | 6.1 ± 1.3Aa | 64.2 ± 12.1Ba | 1.45 ± 0.79 | -0.80 ± 0.79 |

| Vegetation Type | Methane Flux ($mg\ CH_4\text{-}C\ m^{-2}\ d^{-1}$) | |
|---|---|---|
| | Wet Season | Dry Sea: |
| Forested | 6.7 ± 1.0Aa | 47.2 ± 5 |
| Forested (short pole) | 60.4 ± 9.1Ab | 18.8 ± 2 |
| *M. flexuosa* Palm Swamp | 46.7 ± 8.4Ac | 28.3 ± 2 |
| Mixed Palm Swamp | 6.1 ± 1.3Aa | 64.2 ± 1 |

**Figure Captions**

**Figure 1.** Map of the study region and field sites. The colour scale to the right of the map denotes elevation in meters above sea level (m a.s.l.). Tan and brown tones indicate peatland areas.

**Figure 2.** Net diffusive **(a)** methane (CH$_4$) and **(b)** nitrous oxide (N$_2$O) fluxes by vegetation type. Error bars denote standard errors.
* * *
**Figure 1**

[Figure]

**Figure 2**

[Figure]

[Figure]

**8.**

---

## Referee Report (RR1)

**Referee report on "Seasonal variability in methane and nitrous oxide fluxes from tropical peatlands in the Western Amazon basin" by Y. A. Teh et al.**

The revised manuscript by Teh et al. has included the most important comments and suggestions by the two referees. Thus, the quality of the manuscript was substantially improved and it should be appropriate for publication in *Biogeosciences* from my point of view.

However, I still have one concern regarding the process of "ebullition-driven $CH_4$ uptake". You should not treat this as an independent process from ebullition as in l. 433 – 434 where you state that there was no evidence of ebullition but you measured ebullition-driven uptake. Logically, this doesn't make sense as there can only be ebullition-driven uptake if there is ebullition. For the forested vegetation, no ebullition without a subsequent $CH_4$ uptake was measured. Doesn't this simply mean that the $CH_4$ oxidation potential was very high at that site? This might be explained by the water levels, which were lowest for the forested site during the dry season, and the generally high $CH_4$ oxidation potentials of forest soils. These points could be pointed out more precisely.

---

## Author Response (AR2)

REFEREE 1

1. I have now read the revised manuscript by Teh et al. I would like to commend the authors for the extended way they addressed the reviewers concerns and the careful revision that they made. I have one minor issue left, which may not be solvable. While the map in Fig. 1 is of much better quality, it does not show the strata that were used for sampling. Is it not possible to show this? The present figure is a simple elevation map. Of course there will be a strong correlation with elevation, but is it as simple as the colour-coding suggests? If the map is correct this would mean that all of Iquitos is located on peats. This may be true (I have never been there), but seems unlikely for a town of that size.

AUTHOR RESPONSE: Please accept our apologies for misinterpreting your prior question and not providing all the information you requested in your earlier review. Unfortunately, we do not have detailed ground-truthed maps delineating the peatlands sampled in this study and are unable to provide the information that the referee has requested. While prior investigators have estimated regional vegetation distributions using a supervised image classification approach, there are some significant methodological limitations associated with this method, not least of which is that even multiple satellite products are unable to distinguish among all the major vegetation types in the region, including the vegetation types studied here (Lahteenoja et al., 2012;Draper et al., 2014). Moreover, large portions of the satellite data for PMFB have not been adequately ground-truthed for vegetation type or peat characteristics (e.g. peat depth, bulk density) (Lahteenoja et al., 2012;Draper et al., 2014). We have revised the caption for Figure 1 to better reflect what the figure does in fact represent.

REFEREE 2

2. The revised manuscript by Teh et al. has included the most important comments and suggestions by the two referees. Thus, the quality of the manuscript was substantially improved and it should be appropriate for publication in *Biogeosciences* from my point of view. However, I still have one concern regarding the process of "ebullition-driven CH4 uptake". You should not treat this as an independent process from ebullition as in l. 433 – 434 where you state that there was no evidence of ebullition but you measured ebullition-driven uptake. Logically, this doesn't make sense as there can only be ebullition-driven uptake if there is ebullition. For the forested vegetation, no ebullition without a subsequent CH4 uptake was measured. Doesn't this simply mean that the CH4 oxidation potential was very high at that site? This might be explained by the water levels, which were lowest for the forested site during the dry season, and the generally high CH4 oxidation potentials of forest soils. These points could be pointed out more precisely.

AUTHOR RESPONSE: Thank you for these perceptive remarks. The referee is correct regarding lines 433-434; the second revision of the manuscript has incorporated the referee's suggestion. We have re-written this sentence to make our statements more logical (see lines 441-446 in the second revision of the text). We have also revised the remainder of the text in order to refer to observations where a net emission was observed as "net ebullition". The reasoning behind this is that both "net ebullition" and "ebullition-derived uptake" ultimately stem from a bubble (ebullition) event, and the subtle change in terminology we have employed in the second revision will reflect a more precise use of language (line 24, lines 304-309, line 341, line 346, line 357-361, line 439-446, line 507-508, lines 536-537, Table 2).

With respect to the referee's remark about CH4 oxidation; while we do agree that the data may suggest higher rates of oxidation in forested and forested (short pole) vegetation types, we cannot discount the possibility that the differences in net diffusive flux and ebullition may be due to differences in production among sites and/or transport/bubble formation processes in the soil. We have subtly revised the discussion section to acknowledge this point (line 563).

REFERENCES

[revised manuscript text omitted]